# $\mathcal{X}^2$-DFD: A framework for e$\mathcal{X}$plainable and e$\mathcal{X}$tendable Deepfake Detection

## Abstract

Detecting deepfakes (*i.e.*, AI-generated content with malicious intent) has become an important task. Most existing detection methods provide only real/fake predictions without offering human-comprehensible explanations. Recent studies leveraging multimodal large-language models (MLLMs) for deepfake detection have shown improvements in explainability. However, the performance of pre-trained MLLMs (*e.g.*, LLaVA) remains limited due to a lack of understanding of their capabilities for this task and strategies to enhance them. In this work, we empirically assess the strengths and weaknesses of MLLMs specifically in deepfake detection via forgery-related feature analysis. Building on these assessments, we propose a novel framework called $\mathcal{X}^2$-DFD, consisting of three core modules. The first module, *Model Feature Assessment (MFA)*, measures the detection capabilities of forgery-related features intrinsic to MLLMs, and gives a descending ranking of these features. The second module, *Strong Feature Strengthening (SFS)*, enhances the detection and explanation capabilities by fine-tuning the MLLM on a dataset constructed based on the top-ranked features. The third module, *Weak Feature Supplementing (WFS)*, improves the fine-tuned MLLM's capabilities on lower-ranked features by integrating external dedicated deepfake detectors. To verify the effectiveness of this framework, we further present a practical implementation, where an automated forger-related feature generation, evaluation, and ranking procedure is designed for *MFA* module; an automated generation procedure of the fine-tuning dataset containing real and fake images with explanations based on top-ranked features is developed for *SFS* model; an external conventional deepfake detector focusing on blending artifact, which corresponds to a low detection capability in the pre-trained MLLM, is integrated for *WFS* module. Experimental results show that the proposed implementation enhances overall detection performance compared to pre-trained MLLMs, while providing more convincing explanations. More encouragingly, our framework is designed to be plug-and-play, allowing it to seamlessly integrate with more advanced MLLMs and external detectors, leading to continual improvement and extension to face the challenges of rapidly evolving deepfake technologies.

## 1 Introduction

Current generative AI technologies have enabled easy manipulation of facial identities, with many applications such as filmmaking and entertainment (Pei et al., 2024). However, these technologies can also be misused to create *deepfakes*[1] for malicious purposes, including violating personal privacy, spreading misinformation, and eroding trust in digital media. Hence, there is a pressing need to establish a reliable and robust system for detecting deepfakes. In recent years, numerous deepfake detection methods have been proposed (Li, 2018; Liu et al., 2021a; Zhao et al., 2021a; Li et al., 2020a; Chen et al., 2022; Shiohara & Yamasaki, 2022; Yan et al., 2023c;a), with most focusing on addressing the generalization issue that arises from the discrepancies between training and testing data distributions. Despite improvements in generalization performance, these methods typically only output a probability indicating whether a given input is AI-generated, without providing intuitive and convincing explanations behind the prediction. This lack of reliable explanations confuses

---

[1]The term "deepfake" used here refers explicitly to **face** forgery images or videos. Full (natural) image synthesis is not strictly within our scope.

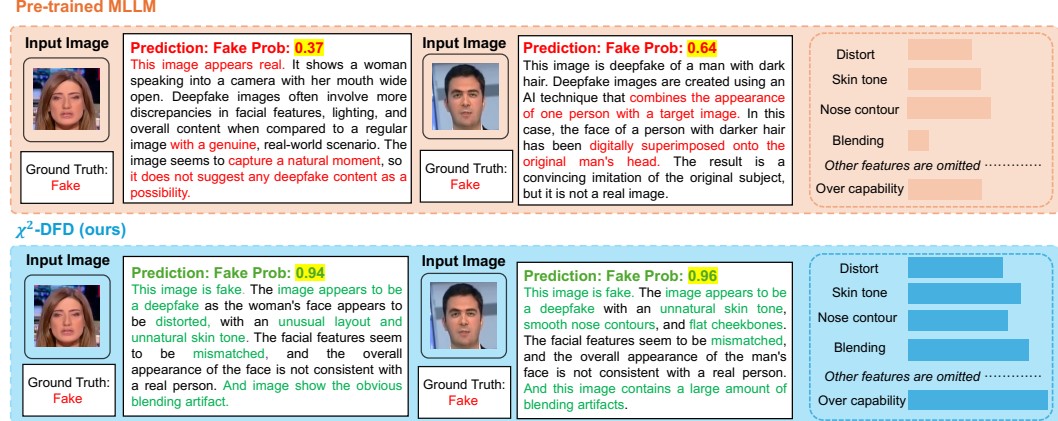

Figure 1: Illustration of the differences between the pre-trained MLLM and ours in deepfake detection. We demonstrate the prediction, explanation, and capability assessment results (see the right column, where each index corresponds to a forgery-related feature) for comparison. Our framework enhances both the detection capability and explanation of the pre-trained MLLM by improving strong features (*e.g.*, skin tone and nose contour) and supplementing weak features (*e.g.*, Blending).

users about why it is deemed fake. In some critical scenarios like incorporating the detection result into judicial evidence, explanations are underlying essential.

Multimodal Large Language Models (MLLMs) have shown remarkable potential in many research areas (Yin et al., 2023). Given their advanced vision-language integration capabilities, MLLMs hold promise for addressing the explainability gap. A few recent efforts (Jia et al., 2024; Shi et al., 2024) have explored leveraging pre-trained MLLMs to obtain explainability for deepfake detection. However, our preliminary studies reveal limitations in pre-trained MLLMs, primarily due to an insufficient understanding of their capabilities specific to deepfake detection and a lack of effective strategies to enhance their performance. Specifically, we investigate the discrimination of several forgery-related features (*e.g.*, blending, lighting) in the pre-trained MLLM (*e.g.*, LLaVA), and find significant differences. As shown in Fig. 1, we see that some features exhibit strong discriminative capability for deepfake detection, while others do not. This discrepancy may explain the limited detection performance of the pre-trained MLLM, as well as its unreasonable explanations.

Inspired by the above investigation, we propose $\mathcal{X}^2$-**DFD**, a novel framework that utilizes MLLMs for e$\mathcal{X}$plainable and e$\mathcal{X}$tendable **D**eep**F**ake **D**etection. The proposed $\mathcal{X}^2$-DFD operates through three core modules. **First**, the *Model Feature Assessment (MFA) Module* aims to assess the intrinsic capability of the pre-trained MLLMs in deepfake detection. We provide a quantified assessment of the discriminative capability for detection of each forgery-related feature, leading to a descending ranking of all candidate features. **Second**, the *Strong Feature Strengthening (SFS) Module* aims to improve the overall detection performance of the model by fully leveraging strong features (*i.e.*, top-ranked intrinsic capabilities) for model fine-tuning. **Third**, the *Weak Feature Supplementing (WFS) Module* aims to supplement the weak intrinsic capabilities of the model by leveraging the strength of external dedicated detectors (EDDs) for weak features (*i.e.*, low-ranked intrinsic capabilities). Encouragingly, the modular-based design of the proposed $\mathcal{X}^2$-**DFD** framework enables seamless integration with future MLLMs and EDDs as their capabilities evolve.

Our main contributions are threefold. **1)** Studying the intrinsic capabilities of MLLMs for deepfake detection: To our knowledge, we are the first to systematically assess the inherent capabilities of MLLMs specifically in deepfake detection. We reveal that MLLMs have varying discriminating capabilities on different forgery features. **2)** Enhancing MLLMs' explainability through designed fine-tuning: Based on the identified strengths of MLLMs, we fine-tune them to generate explanations grounded in their most "familiar" forgery features and abandon those they "unfamiliar" with, thereby improving their ability to accurately detect and convincingly explain deepfakes. **3)** For areas where MLLMs show limitations, we integrate EDDs to supplement the model's weakness. This allows us to leverage the strength of both MLLMs and EDDs for a better detection system.

## 2 RELATED WORK

**Conventional Deepfake Detection** Early detection methods typically focus on performing feature engineering to mine a manual feature such as eye blinking frequency (Li et al., 2018), warping artifacts (Li, 2018), headpose (Yang et al., 2019), and *etc*. Recent conventional deepfake detectors mainly focus on dealing with the issue of generalization (Yan et al., 2023d), where the distribution of training and testing data varies. Until now, there have developed novel solutions from different directions: constructing pseudo-fake samples to capture the blending clues (Li, 2018; Li et al., 2020a; Shiohara et al., 2023; Zhao et al., 2021b), learning spatial-frequency anomalies (Gu et al., 2022; Liu et al., 2021a; Luo et al., 2021a; Qian et al., 2020a), focusing on the ID inconsistency clues between fake and corresponding real (Dong et al., 2023), performing disentanglement learning to learn the forgery-related features (Yan et al., 2023b; Yang et al., 2021), performing reconstruction learning to learn the general forgery clues (Cao et al., 2022b; Wang & Deng, 2021), locating the spatial-temporal inconsistency (Haliassos et al., 2021; Wang et al., 2023; Zheng et al., 2021a; Yan et al., 2024b), and *etc*. Most of these methods improve the generalization ability compared to the early detection methods. However, these methods can provide only real/fake predictions without giving detailed explanations behind the predictions. The lack of convincing and human-comprehensible explanations might confuse users about why it is deemed fake.

**Deepfake Detection via Multimodal Large Language Model** Vision and language are the two important signals for human perception, and visual-language multimodal learning has thus drawn a lot of attention in the AI community. Recently, the LLaVA series (Liu et al., 2023b; 2024; 2023a) have explored a simple and effective approach for visual-language multimodal modeling. In the field of deepfake detection, (Jia et al., 2024; Shi et al., 2024) have investigated the potential of prompt engineering in face forgery analysis and proposed that existing MLLMs show better explainability than previous conventional deepfake detectors. In addition, Li et al. (2024b) probe MLLMs for explainable fake image detection by presenting a labeled multimodal database for fine-tuning. More recently, (Zhang et al., 2024) proposed using pairs of human-generated visual questions answering (VQA) to construct the fine-tuning dataset, but manually creating detailed annotations can be costly. Another just-released work (Huang et al., 2024) proposes an automatic approach using GPT-4o(Achiam et al., 2023) to generate annotations and train MLLM with the resulting VQA pairs. However, a new critical question was then raised: *Can MLLMs (e.g., LLaVa) fully comprehend the fake clues identified by GPT-4o?* It is reasonable to believe that there remains a capability gap between LLaVa and GPT-4o. For this reason, we find that existing works lacking in understanding the limitations of capability and then find ways to enhance the strengths and augment the limitations.

## 3 INVESTIGATION OF PRE-TRAINED MLLMS' CAPABILITY IN DEEPFAKE DETECTION

### 3.1 EVALUATION SETUP

**Model.** We choose the mainstream MLLM, *i.e.*, LLaVA (Liu et al., 2023b) as the implementation instance of the pre-trained MLLMs. Additionally, we choose one classical external dedicated detector (EDD), Xception (Chollet, 2017), as a baseline model for comparison.

**Dataset.** We evaluate the models on several widely-used deepfake datasets, including the Deepfake Detection Challenge (DFDC) (Dolhansky et al., 2020), the preview version of DFDC (DFDCP) (Dolhansky et al., 2019), DeepfakeDetection (DFD) (Deepfakedetection., 2021), Celeb-DF-v2 (CDF-v2) (Li et al., 2020b), as well as the newly released DF40 dataset (Yan et al., 2024a). The DF40 dataset incorporates a variety of forgery techniques, including Facedancer (Rosberg et al., 2022), FSGAN (Nirkin et al., 2019), inSwap (Sangwan, 2020), e4s (Li et al., 2024a), Simswap (Chen et al., 2020), and Uniface (Zhou et al., 2023), providing a comprehensive foundation for evaluating overall detection performance.

**Evaluation Metrics.** We use the Area Under the Curve (AUC) as the primary evaluation metric, enabling us to assess the model's ability to distinguish between real and fake images across the whole dataset. In this section, we use the frame-level AUC for evaluation. For individual feature discrimination, we focus on forgery-related features such as "Is the face layout unnatural?" with responses of either "yes" or "no." The proportions of "yes" and "no" answers for real and fake

images are calculated as follows, with the ranking score $S^{(q)}$ defined based on the balanced accuracy of the responses:

$$S^{(q)} = \frac{1}{2} \left( \frac{Y_{\text{real}}^{(q)}}{Y_{\text{real}}^{(q)} + N_{\text{real}}^{(q)}} + \frac{N_{\text{fake}}^{(q)}}{Y_{\text{fake}}^{(q)} + N_{\text{fake}}^{(q)}} \right). \tag{1}$$

Here, $Y_{\text{real}}^{(q)}$ and $Y_{\text{fake}}^{(q)}$ denote the number of "yes" answers, while $N_{\text{real}}^{(q)}$ and $N_{\text{fake}}^{(q)}$ represent the number of "no" answers for real and fake, respectively. This formulation ensures that both true positive and true negative rates are considered, providing a balanced measure of feature discrimination.

## 3.2 EVALUATION OF THE OVERALL DETECTION PERFORMANCE

The comparison between LLaVA (Liu et al., 2023b) and Xception (Chollet, 2017) highlights a notable performance gap. Results in Fig. 2 (left) indicate that the average AUC for LLaVA is 63.7%, while Xception achieves 75.8%, showing a notable gap of 12.1% points. This suggests that, while the LLaVA has certain zero-shot capabilities in other tasks such as (general) image classification, it is still not as strong as the EDD in detecting deepfakes.

However, LLaVA shows strong detection abilities in specific methods (*e.g.*, e4s), sometimes even surpassing Xception (see Fig. 2 (left)). This motivates us to further investigate its intrinsic detection capabilities, and understand the model's "strengths and weaknesses" in deepfake detection. Below, we provide a detailed investigation of the discrimination of each forgery-related feature.

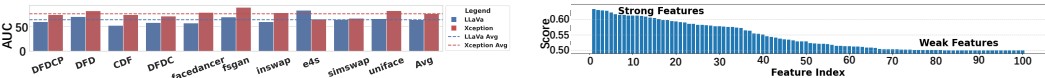

Figure 2: (Left) AUC comparison between (zero-shot) LLaVA (blue) and Xception (red) for deepfake detection across different datasets; (Right) Balance accuracy score for individual feature discrimination, with Strong features in the top-left corner and Weak features in the bottom-right corner based on discrimination scores. Full questions/features are provided in the Appendix A.1.

## 3.3 INVESTIGATION OF INDIVIDUAL FEATURE'S DISCRIMINATION

As shown in the top row of Fig. 4, our implementation encompasses three consecutive steps.

**Step 1: Question Generation.** For each candidate forgery-related feature, we formulate a corresponding interrogative statement. For instance, the feature "*blurry*" is transformed into the question "*Is the image blurry?*". Recognizing that the candidate features are not pre-specified by developers, we employ a Large Language Model (LLM), *i.e.*, GPT-4o, to automatically generate a comprehensive list of $N_q$ questions, denoted as $Q_i^k$ for $k \in \{1, \ldots, N_q\}$. These questions target key forgery indicators, including but not limited to lighting anomalies, unnatural facial expressions, and mismatched textures, which are critical for identifying deepfakes.

**Step 2: Question Evaluation.** Referring to Section 3.2, each generated question is paired with an image from the assessment dataset to form a prompt for constructing the fine-tuning dataset. The model responds with a binary output ("yes" or "no") based on its interpretation of the image in relation to the question. These responses are aggregated into a confusion matrix for each question, thereby quantifying the detection capability of the associated forgery-related features. Mathematically, for each question $Q_i^k$ and image $x_j$, the MLLM produces:

$$R_{i,j}^k = \mathcal{M}_{\text{base}}(Q_i^k, x_j), \tag{2}$$

where $R_{i,j}^k \in \{\text{yes}, \text{no}\}$, representing the model's response for each image-question pair.

**Step 3: Question Ranking.** According to the accuracy of all candidate questions, we obtain a descending ranking of questions, *i.e.*, the ranking of forgery-related features. This ranking allows us to quantify how well each feature contributes to distinguishing between real and fake images. Specifically, the accuracy of each question is computed by evaluating the proportion of correct responses

across the dataset. Specifically, for each question $Q_i^k$, We calculate the true positive rate (TPR) and true negative rate (TNR), then take their average to obtain the Balanced Accuracy, as follows:

$$\text{Balanced Accuracy}_i^k = \frac{1}{2}\left(\frac{\text{TP}_i^k}{\text{TP}_i^k + \text{FN}_i^k} + \frac{\text{TN}_i^k}{\text{TN}_i^k + \text{FP}_i^k}\right), \tag{3}$$

where: $\text{TP}_i^k$ denotes True Positives for question $Q_i^k$, $\text{TN}_i^k$ the True Negatives for question $Q_i^k$, $\text{FP}_i^k$ the False Positives for question $Q_i^k$, and $\text{FN}_i^k$ the False Negatives for question $Q_i^k$.

Subsequently, questions are ranked in descending order based on their balance accuracy scores, thereby prioritizing forgery features that effectively discriminate between real and fake images.

**Strong Features.** Strong features typically involve *semantic-level facial structural or appearance anomalies*. As shown in the strong feature section of Fig. 2 (right), which primarily includes facial irregularities such as unusual facial layouts (*e.g.*, *Rank 9, 11, 17*) or distorted facial features (*e.g.*, *Rank 3, 4, 14*), *e.g.*, the nose, eyes, or mouth. Since the pre-trained MLLM is good at extracting and utilizing these features for detection, it can provide a more reliable and accurate explanation.

**Weak Features.** Weak features typically involve *fine-grained, low-level textures*, such as blending anomalies. As shown in Fig. 2 (right), these weak features are primarily subtle details related to texture, reflection, shadow, and blending. Examples of texture issues include rough or overly smooth surfaces (*e.g.*, *Rank 68, 77, 83*). Furthermore, inconsistencies in lighting and shadows (*e.g.*, *Rank 85, 86, 90, 96*) and blending artifacts on the face (*e.g.*, *Rank 54, 84, 88*) are also prominent. Since these signal-level anomalies are challenging for pre-trained MLLMs to detect, the pre-trained MLLM is likely to struggle in reliably distinguishing between real and manipulated content when relying on these weak features for detection and explanation.

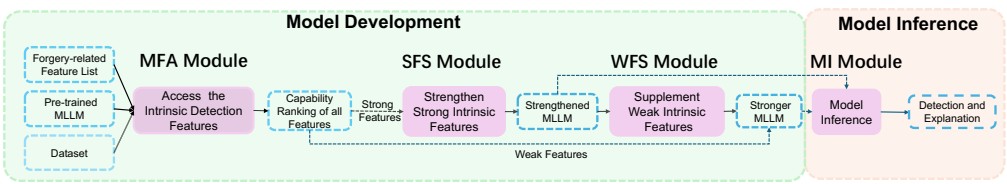

Figure 3: High-level pipeline of the e$\mathcal{X}$plainable and e$\mathcal{X}$tendable **D**eep**F**ake **D**etection ($\mathcal{X}^2$-DFD) framework, which contains three core modules: MFA, SFS, and WFS for model development, and one MI module for inference. Detailed text can be seen in Sec. 4.1.

## 4 OUR METHODOLOGY

### 4.1 A GENERAL FRAMEWORK FOR DEEPFAKE DETECTION

As illustrated in Fig. 3, we design a novel framework for the deepfake detection task based on MLLMs, called e$\mathcal{X}$plainable and e$\mathcal{X}$tendable **D**eep**F**ake **D**etection ($\mathcal{X}^2$-DFD). Our framework contains two main parts: model development (the core one) and model deployment. For model development, the definition and role of each module are demonstrated as follows:

- **Model Feature Assessment Module** (MFA Module): Given an assessment dataset and a candidate list of forgery-related features, this module assesses the inherent detection capability of each feature in the initial pre-trained MLLM. It outputs a capability ranking of all discriminative features in detecting deepfakes.

- **Strong Feature Strengthening Module** (SFS Module): According to the capability ranking, this module aims to strengthen good intrinsic capabilities to improve the overall detection performance of the initial pre-trained MLLM, and outputs a strengthened MLLM.

- **Weak Feature Supplementing Module** (WFS Module): Based on the capability ranking and the strengthened MLLM, this module aims to supplement weak intrinsic capabilities, and outputs a stronger MLLM.

**Model Inference (MI) Module** is implemented to output the predictions and detailed explanations. Specifically, this module aims to deploy our final MLLM for inference purposes, $i.e.$, detecting deepfakes (providing real/fake prediction) and explaining deepfakes (giving detailed reasons behind the prediction).

**Future extension to an (automatic) close-loop framework**: We propose adding a user feedback loop to the MFA module. This extension would allow for continuous model improvement by iteratively incorporating user feedback, which would adjust the model's focus on certain features and further refine its performance.

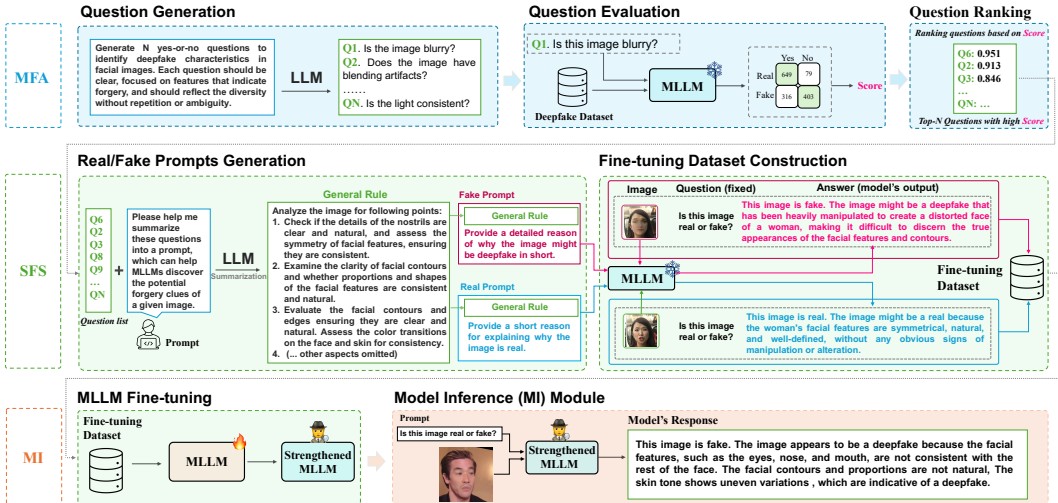

Figure 4: The discrete implementation of the proposed framework, where an automated forger-related feature generation, evaluation, and ranking procedure is designed for *MFA* module; an automated generation procedure of the fine-tuning dataset containing real and fake images with explanations based on strong features is developed for *SFS* model. The implementation of *WFS* module can be seen in Fig. 5. Detailed text is in Sec. 4.2.

## 4.2 An Implementation of the Detection Framework

In this subsection, we delineate a **concrete implementation** of the proposed $\mathcal{X}^2$-DFD framework, utilizing a pre-trained Multimodal Large Language Model (MLLM), $e.g.$, LLaVA. The architecture of the implementation is depicted in Fig. 4. Our implementation is divided into four primary modules: Model Feature Assessment (MFA), Strong Feature Strengthening (SFS), Weak Feature Supplementing (WFS), and Model Inference (MI) Module. We will introduce them below.

### 4.2.1 Implementation of MFA Module

As shown in the top row of Fig. 4, MFA encompasses three consecutive steps: question generation, evaluation, and ranking. The methodology for each of these steps has been explained in detail in the previous section. For more information, please refer to Sec. 3.3. These obtained questions are ranked in descending order based on their accuracy scores, thereby prioritizing forgery-related features that most effectively discriminate between real and fake images. After ranking the generated questions by LLMs, we conduct human verification to ensure the reliability and accuracy of the fake features. Note that most questions are generated well without any obvious errors or irrelevant information.

### 4.2.2 Implementation of SFS Module

The SFS module fine-tunes the MLLM by strengthening its ability to detect features that have been identified as high-performing in the MFA module. This process consists of three key steps:

**Step 1: Real/Fake Prompts Generation.** Leveraging the strong features from the MFA module, we generate specialized prompts to guide the MLLM's focus during the fine-tuning phase. Specifically, we first utilize GPT-4o to summarize these strong features and construct two distinct prompts: one

tailored for real images ($\mathbf{P}_{\text{real}}$) and another for fake images ($\mathbf{P}_{\text{fake}}$). These prompts are formulated as: $\mathbf{P}_{\text{real}} = f(\mathbf{F}_{\text{real}})$, $\mathbf{P}_{\text{fake}} = f(\mathbf{F}_{\text{fake}})$, where $\mathbf{F}_{\text{real}}$ and $\mathbf{F}_{\text{fake}}$ denote the sets of strong features relevant to real and fake images, respectively. Also, $f$ represents any LLMs. Here, we employ GPT-4o for implementation.

**Step 2: Fine-tuning Dataset Construction.** A fine-tuning dataset $D_{ft}$ comprising VQA-style (visual question answering) pairs, which is constructed by pairing each image with the corresponding (real or fake) prompt. Each image is annotated with the specific features it exhibits, and the standardized prompt $\mathbf{P}_{\text{fixed}}$ is defined as: $\mathbf{P}_{\text{fixed}} =$ "Is this image real or fake?" The model's response is structured to begin with a definitive statement—*"This image is real/fake"*—followed by an explanation based on the identified features. Formally, the final answer is represented as: $\mathbf{A}_{\text{final}} =$ "This image is real/fake" $+ \mathbf{A}_{\text{real/fake}}$. Consequently, each VQA-style pair of the fine-tuning dataset $D_{ft}$ is formalized as: $\mathbf{VQA} = (Image, \mathbf{P}_{\text{fixed}}, \mathbf{A}_{\text{final}})$.

**Step 3: MLLM Fine-tuning.** The initial MLLM is fine-tuned using $D_{ft}$. The fine-tuning process involves adjusting the *projector* to accurately associate image artifacts with the corresponding fake labels. Additionally, Low-Rank Adaptation (LoRA) (Hu et al., 2021) is employed to selectively fine-tune a subset of the model's parameters, thereby focusing the model's reasoning on deepfake-specific features while maintaining overall model integrity. The fine-tuning process can be denoted as: $\mathcal{M}_{\text{base}} \xrightarrow{D_{ft}} \mathcal{M}_{\text{fine-tuned}}$, where $\mathcal{M}_{\text{fine-tuned}}$ is the enhanced MLLM with improved deepfake detection capabilities.

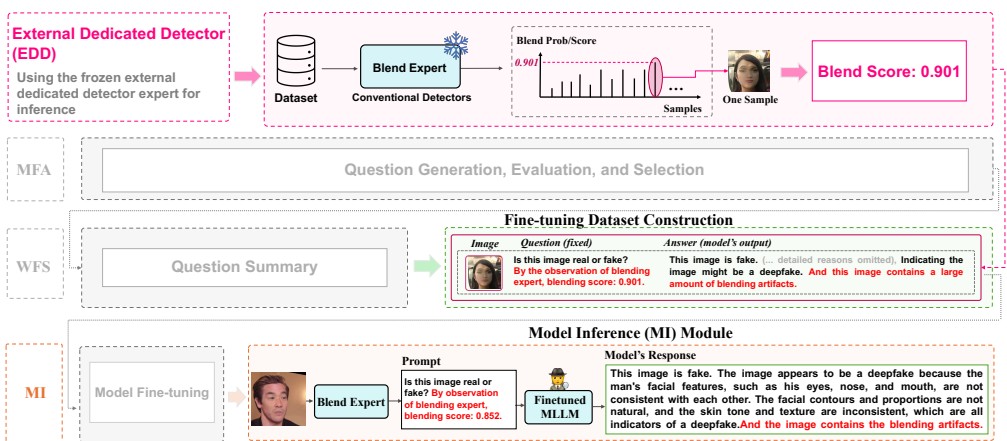

Figure 5: Illustration of the pipeline after adding WFS Module, which enhances MLLM deepfake detection by integrating the external dedicated detector, creating an updated fine-tuning dataset. During inference, the MLLM is enhanced by incorporating information from the external detector.

### 4.2.3 IMPLEMENTATION OF WFS MODULES

The WFS module enhances the MLLM by integrating external deepfake detectors, which are specialized in detecting features where the MLLM shows weakness. The overall pipeline after adding the WFS module is illustrated in Fig. 5. This module follows three steps:

**Step 1: External Detector Invocation.** For features that the MLLM identifies as weak, we deploy an external specialized deepfake detector (*e.g.*, a blending-based detector (Lin et al., 2024)). This external detector processes the input image and generates a prediction. Note that we also employ other EDDs for implementation, and we provide an in-depth analysis for this in Sec. A.2. Specifically, when utilizing a blending detector as an instance of EDD, a blending score $s$ is produced: $s = \sigma(\text{BlendDetector}(x))$, where $x$ denotes the input image, and $\sigma$ denotes the sigmoid function that transforms the logits output of the BlendDetector into the 0-1 range.

**Step 2: Integration of External Detection Results into the Fine-tuning Dataset.**

The blending score $s$ obtained from the external detector is incorporated into the fine-tuning dataset by appending it to the existing prompts. This is done by adding a statement such as: *"By the observation of the blending expert, blending score: $\underline{s}$"* Additionally, based on the score, a corresponding response aligned with the probability is included, as shown in Fig. 5, specifically in the Fine-tuning

Dataset Construction section of the SFS. This integration ensures that the MLLM benefits from both its intrinsic detection capabilities and the specialized insights provided by the EDD.

**Step 3: Integration of External Detection Results into Inference Prompts** During inference, the EDD's results are integrated into the MLLM's prompt-based reasoning process. The detailed description and formulation can be seen in Sec. 4.2.4 below.

### 4.2.4 IMPLEMENTATION OF MI MODULE

Generally, during the inference, the external detector's blending score $s$ is incorporated into the MLLM's prompt-based reasoning process. Specifically, the final output of the model is structured, to begin with a definitive statement—*"This image is real/fake"*—followed by reasoning based on identified visual features. Based on the blending score $s$, the model appends a descriptive statement: $\mathbf{A}_{\text{final}}$ = *"This image is real/fake"* + $\mathbf{A}_{\text{real/fake}}$ + *" And this image contains obvious/minimal blending artifacts."* The model acquires this response pattern through training. This approach ensures that the MLLM effectively leverages EDDs to enhance its detection performance, particularly for features where it initially demonstrated weakness.

## 5 EXPERIMENTS

### 5.1 EXPERIMENTAL SETTINGS

**Datasets.** Following previous works (Yan et al., 2023d;a) We conduct experiments on the following commonly used datasets: FaceForensics++ (FF++) (Rossler et al., 2019), DFDC, DFDCP, DFD, CDF-v2, DFo (Jiang et al., 2020), WDF (Zi et al., 2020), and FFIW (Zhou et al., 2021). In line with the standard deepfake benchmark (Yan et al., 2023d), we use the c23 version of FF++ for training and other datasets for testing (**Protocol-1**). We also evaluate the models on the **just-released** deepfake dataset DF40 (Yan et al., 2024a), which contains many latest SOTA forgery methods on the FF++ domain. We select six face-swapping methods generated from the FF++ domain for cross-manipulation evaluation (**Protocol-2**).

**Implementation Details.** We fine-tune the LLaVA model (Liu et al., 2023b) using the VQA dataset. For the conventional model, we use a blending-based approach proposed in (Lin et al., 2024). Training is conducted on a single NVIDIA 4090 GPU for 1 epoch, with a learning rate of 2e-5, rank set to 16, and alpha, conventionally set to twice the rank, at 32. The batch size is set to 4, and we use a gradient accumulation step of 1. For evaluation metrics, we mainly report both the frame-level and video-level AUC of our results. Other metrics such as Accuracy (Acc.), Equal Error Rate (EER), and Average Precision (AP) are also reported. More details can be seen in the Appendix A.8.

**Compared Baselines.** We compare 24 methods both frame level and video level. In which Xception (Chollet, 2017), Efficient-b4 (Tan & Le, 2019), FWA (Li, 2018), Face X-ray (Li et al., 2020a), RECCE (Cao et al., 2022a), F3-Net (Qian et al., 2020b), SPSL (Liu et al., 2021b), SRM (Luo et al., 2021b), UCF (Yan et al., 2023c), IID (Huang et al., 2023), ICT (Dong et al., 2022), ViT-B (Dosovitskiy et al., 2020), ViT-B (Radford et al., 2021), LSDA (Yan et al., 2023a), PCL+I2G (Zhao et al., 2021c), LipForensics (Haliassos et al., 2021), FTCN (Zheng et al., 2021a), CORE (Ni et al., 2022), SBI (Shiohara & Yamasaki, 2022), UIA-ViT (Zhuang et al., 2022), SLADD (Chen et al., 2022), DCL (Sun et al., 2021), SeeABLE (Larue et al., 2023), and CFM (Luo et al., 2023).

### 5.2 PROTOCOL-1: CROSS-DATASET EVALUATION

In Tab. 1, we compare our method with 24 SOTA detectors via cross-dataset evaluations. The results of other compared baselines are mainly cited from their original papers. Ours consistently outperforms other models across **all** tested scenarios, demonstrating its better detection performance. Our approach excels across both frame-level and video-level evaluations, maintaining superior results when compared to other methods. The table clearly highlights our method's capability to generalize and consistently achieve higher accuracy on cross-dataset tasks.

### 5.3 PROTOCOL-2: CROSS-MANIPULATION EVALUATION

Evaluating our model's performance on cross-manipulation tasks helps assess whether it can handle previously unseen fake types. We use the recently released DF40 dataset (Yan et al., 2024a) for evaluation. Our method generally outperforms other models on average, particularly the e4s,

Table 1: **Protocol-1:** Cross-dataset evaluations with **24 existing detectors**. All detectors are trained on FF++_c23 (Rossler et al., 2019) and evaluated on other datasets. The top two results are highlighted, with the best in bold and the second underlined. '*' indicates our reproductions.

| | | **Frame-Level AUC** | | | | | | | **Video-Level AUC** | | | | |
|---|---|---|---|---|---|---|---|---|---|---|---|---|---|
| Method | Venues | CDF | DFDCP | DFDC | DFD | Avg | Method | Venues | CDF | DFDCP | DFDC | DFD | Avg |
| Xception | CVPR 2017 | 73.7 | 73.7 | 70.8 | 81.6 | 75.0 | Xception | CVPR 2017 | 81.6 | 74.2 | 73.2 | 89.6 | 79.7 |
| FWA | CVPRW 2018 | 66.8 | 63.7 | 61.3 | 74.0 | 66.5 | PCL+I2G | ICCV 2021 | 90.0 | 74.4 | 67.5 | - | - |
| Efficient-b4 | ICML 2019 | 74.9 | 72.8 | 69.6 | 81.5 | 74.7 | LipForensics | CVPR 2021 | 82.4 | - | 73.5 | - | - |
| Face X-ray | CVPR 2020 | 67.9 | 69.4 | 63.3 | 76.7 | 69.3 | FTCN | ICCV 2021 | 86.9 | 74.0 | 71.0 | 94.4 | 81.6 |
| F3-Net | ECCV 2020 | 77.0 | 77.2 | 72.8 | 82.3 | 77.3 | ViT-B (CLIP) | ICML 2021 | 88.4 | 82.5 | 76.1 | 90.0 | 84.3 |
| SPSL | CVPR 2021 | 76.5 | 74.1 | 70.1 | 81.2 | 75.5 | CORE | CVPRW 2022 | 80.9 | 72.0 | 72.1 | 88.2 | 78.3 |
| SRM | CVPR 2021 | 75.5 | 74.1 | 70.0 | 81.2 | 75.2 | SBIs* | CVPR 2022 | 90.6 | 87.7 | 75.2 | 88.2 | 85.4 |
| ViT-B (IN21k) | ICLR 2021 | 75.0 | 75.6 | 73.4 | 86.4 | 77.6 | UIA-ViT | ECCV 2022 | 82.4 | 75.8 | - | 94.7 | - |
| ViT-B (CLIP) | ICML 2021 | 81.7 | 80.2 | 73.5 | 86.6 | 80.5 | SLADD* | CVPR 2022 | 79.7 | - | 77.2 | - | - |
| RECCE | CVPR 2022 | 73.2 | 74.2 | 71.3 | 81.8 | 75.1 | DCL | AAAI 2022 | 88.2 | 76.9 | 75.0 | 92.1 | 83.1 |
| IID | CVPR 2023 | 83.8 | 81.2 | - | - | - | SeeABLE | ICCV 2023 | 87.3 | 86.3 | 75.9 | - | - |
| ICT | CVPR 2023 | 85.7 | - | - | 84.1 | - | CFM | TIFS 2023 | 89.7 | 80.2 | 70.6 | 95.2 | 83.9 |
| UCF | ICCV 2023 | 73.5 | 73.5 | 70.2 | 79.8 | 74.3 | UCF | ICCV 2023 | 83.7 | 74.2 | 77.0 | 86.7 | 80.4 |
| LSDA | CVPR 2024 | 83.0 | 81.5 | 73.6 | 88.0 | 81.5 | LSDA | CVPR 2024 | 89.8 | 81.2 | 73.5 | 95.6 | 85.0 |
| **Ours** | - | **90.3** (+4.6%) | **89.7** (+8.2%) | **83.5** (+9.9%) | **92.5** (+4.5%) | **89.0** (+7.5%) | **Ours** | - | **95.5** (+5.5%) | **91.2** (+4.9%) | **85.3** (+8.3%) | **95.7** (+0.1%) | **91.9** (+6.9%) |

Inswap, and SimSwap methods (see Tab. 2). This shows that our method effectively learns more generalizable features for detection, even against the latest techniques.

Table 2: **Protocol-2:** Cross-manipulation evaluations within the FF++ domain (frame-level AUC only). We leverage the DF40 dataset (Yan et al., 2024a) and select six representative face-swapping methods generated within the FF++ domain, keeping the data domain unchanged. The top two results are highlighted, with the best result shown in bold and the second-best underlined.

| Method | Venues | uniface | e4s | facedancer | fsgan | inswap | simswap | Avg. |
|---|---|---|---|---|---|---|---|---|
| RECCE (Cao et al., 2022a) | CVPR 2022 | 84.2 | 65.2 | 78.3 | 88.4 | 79.5 | 73.0 | 78.1 |
| SBI (Shiohara & Yamasaki, 2022) | CVPR 2022 | 64.4 | 69.0 | 44.7 | 87.9 | 63.3 | 56.8 | 64.4 |
| CORE (Ni et al., 2022) | CVPRW 2022 | 81.7 | 63.4 | 71.7 | 91.1 | 79.4 | 69.3 | 76.1 |
| IID (Huang et al., 2023) | CVPR 2023 | 79.5 | 71.0 | 79.0 | 86.4 | 74.4 | 64.0 | 75.7 |
| UCF (Yan et al., 2023c) | ICCV 2023 | 78.7 | 69.2 | 80.0 | 88.1 | 76.8 | 64.9 | 77.5 |
| LSDA (Yan et al., 2023a) | CVPR 2024 | 85.4 | 68.4 | 75.9 | 83.2 | 81.0 | 72.7 | 77.8 |
| CDFA (Lin et al., 2024) | ECCV 2024 | 76.5 | 67.4 | 75.4 | 84.8 | 72.0 | 76.1 | 75.9 |
| ProgressiveDet (Cheng et al., 2024) | NeurIPS 2024 | 84.5 | 71.0 | 73.6 | 86.5 | 78.8 | 77.8 | 78.7 |
| **Ours** | - | 85.2 | **91.2** | **83.8** | 89.9 | 78.4 | **84.9** | **85.6** |

## 5.4 ABLATION STUDY

In this section, we aim to evaluate the effectiveness of each component proposed in our framework from both detection ability and feature capability aspects.

**Detection Ability.** We evaluate the generalization performance in cross-dataset evaluation scenarios. The ablations involve the following variants. **variant-1**: Baseline (Pre-trained MLLM), which is an initial LLaVA without any feature strengthening or supplementing; **variant-2**: *without* SFS; **variant-3**: *with* SFS; **variant-4**: EDD only, where we use the trained (Lin et al., 2024) for inference; **variant-5**: Ours, which is our final framework with all MFA, SFS, and WFS modules implemented. Results in Tab. 3 demonstrate a clear improvement in AUC, AP, and EER when both SFS and WFS modules are applied, confirming the importance of combining feature strengthening and supplementation for optimal deepfake detection performance.

**Feature Capability.** We also conduct a comparative study of feature capabilities before and after feature strengthening. As shown in Fig. 6, most feature capabilities are significantly enhanced following the application of strong feature strengthening. Notably, even some of the weaker features saw improvements after the enhancement process. A more detailed breakdown and analysis of these improvements are provided in the Appendix A.1.

Table 3: Ablation study regarding the effectiveness of each proposed module via cross-dataset evaluations. The results show an incremental benefit in each module.

| Method | CDF | | | DFD | | | DFDC | | | Simswap | | | Uniface | | | Avg | | |
|---|---|---|---|---|---|---|---|---|---|---|---|---|---|---|---|---|---|---|
| | AUC | AP | EER | AUC | AP | EER | AUC | AP | EER | AUC | AP | EER | AUC | AP | EER | AUC | AP | EER |
| Pre-traind MLLM | 52.1 | 68.2 | 48.7 | 69.8 | 95.2 | 36.4 | 57.8 | 59.9 | 44.6 | 64.0 | 64.1 | 40.4 | 65.5 | 65.6 | 39.0 | 61.8 | 70.6 | 41.8 |
| no SFS | 79.0 | 88.3 | 28.9 | 88.9 | 98.7 | 18.0 | 77.8 | 81.9 | 28.9 | 82.0 | 84.0 | 25.9 | 82.3 | 84.8 | 25.2 | 82.0 | 87.3 | 25.6 |
| with SFS | 83.2 | 90.5 | 24.6 | 91.4 | 99.0 | 15.8 | 79.2 | 82.1 | 27.6 | 83.3 | 85.0 | 24.8 | 84.5 | **86.2** | **22.4** | 84.9 | 88.5 | 23.0 |
| EDD only | 87.9 | 93.6 | 20.5 | 90.9 | 98.9 | 17.6 | 83.5 | **86.1** | 24.8 | 76.0 | 74.2 | 29.8 | 76.5 | 75.1 | 29.8 | 83.0 | 85.6 | 24.5 |
| SFS + WFS | **90.3** | **94.8** | **18.4** | **92.5** | **99.1** | **15.0** | 83.5 | 85.9 | 24.5 | **84.9** | **85.7** | 23.3 | **85.2** | 85.9 | 22.6 | **87.3** | **90.3** | **20.8** |

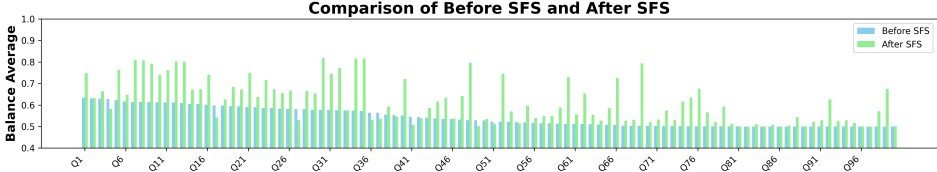

Figure 6: Comparison of feature capability before and after SFS. After adding the external detector to supplement the MLLM, the model's feature capabilities (almost all) can be further improved.

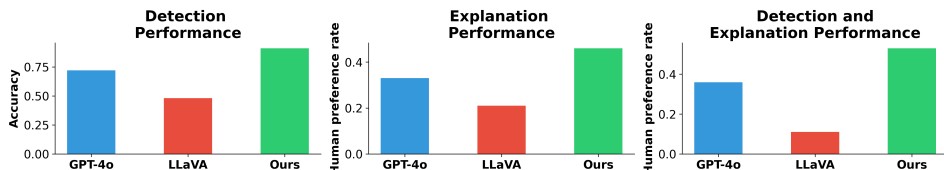

Figure 7: Comparison of detection and explanation performance across GPT-4o, LLaVA, and Ours

## 5.5 HUMAN EVALUATION

We conduct a human evaluation by sampling real and fake images from existing datasets, comparing GPT4o, utilizing the best prompt from (Jia et al., 2024), LLaVa 7B (Pre-trained MLLM) (Liu et al., 2023b), and our model (developed from the Pre-trained LLaVa 7B). Participants evaluate the models across three metrics: detection accuracy, explanation preference, and overall preference for detection and explanations. In all aspects, our model demonstrated superior performance, excelling in both accuracy and human preference for explanations. Further details on the setting description, testing procedure, and analysis can be found in Appendix A.4.

## 6 CONCLUSION

In this paper, we propose $\mathcal{X}^2$-DFD, a novel framework that harnesses the power of Multimodal Large Language Models (MLLMs) for explainable and extendable deepfake detection. For the first time, we systematically evaluate the intrinsic capabilities of the pre-trained MLLMs, revealing their varying effectiveness across different forgery-related features. Inspired by this, we implement a targeted fine-tuning strategy, which has largely improved the explainability of the MLLMs, specifically capitalizing on their strengths. Furthermore, by integrating external deepfake detectors (EDDs), we design a novel framework to combine the complementary advantages of both MLLMs and conventional detectors for better detection and explanation. In the future, we plan to implement our framework in an automated, iterative system that will enable continuous updates based on collected feedback. We hope our work can inspire future advancements in leveraging MLLMs for a better deepfake detection system.

**Content Structure of the Appendix.** Due to limited page content, we put other important analyses and experiments into the Appendix. Specifically, in our appendix, we provide detailed information on the weak Feature Supplementing analysis (A.2), robustness evaluations (A.3), in-domain FF++ test results (A.6), experiments on various LLMs/MLLMs (A.5), and sample demonstrations (A.9).

**Ethics & Reproducibility statements.** All facial images used are from publicly available datasets with proper citations, ensuring no violation of personal privacy. Essential implementation details are in the Appendix, and we will release the code upon acceptance.

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

## A APPENDIX

Due to space constraints, we have included additional important content in the supplementary materials. Below is a brief outline of the supplementary content to facilitate readers easily locate the relevant sections:

- Appendix A.1: Feature Assessment Analysis
- Appendix A.2: Feature Supplementing Analysis
- Appendix A.3: Evaluation of Robustness Against Unseen Perturbations
- Appendix A.4: Human Study Details and Analysis
- Appendix A.5: Experiments on different LLMs/MLLMs
- Appendix A.6: In-domain results in the FF++ dataset
- Appendix A.7: Finding in MLLMs
- Appendix A.8: Implementation Details
- Appendix A.9: Sample Showing

### A.1 FEATURE ASSESSMENT ANALYSIS

**Feature Score.** The scores for different forgery-related features are presented, where Tab. 13 highlights the top 50 strong features, and Tab. 14 shows the 50 weakest features based on their scores.

**Does the model know these features are related to deepfake?**

We used a series of questions to query the model, applying simple prompt augmentation with the feature-related questions mentioned above. A "yes" indicates the model knows these features are related to deepfake detection, while a "no" indicates the model does not. Detailed results are shown in Tab. 11 and Tab. 12.

### A.2 FEATURE SUPPLEMENTING ANALYSIS

In addition to the used blending model Lin et al. (2024), we also try other instances to implement the EDDs in the WCS module of our framework, each targeting specific types of artifacts, where SRI focusing on the generative artifacts by deep nets, F3-Net focusing on the frequency-level anomalies,

and SBI and CFDA focusing on the blending boundaries. Based on these empirical attempts, We summarize the **general criteria** under which conditions the selected EDD instance can be used in our framework. Specifically, **the integrated EDD instance should meet the following criteria:**

- **Criteria-1**: Each EDD instance should focus on only one type of feature that is positively correlated with fake;
- **Criteria-2**: The score given by the EDD instance can accurately reflect the characteristics of the corresponding feature;
- **Criteria-3**: The data distribution of this feature in the dataset is relatively uniform.

Below, we show a detailed illustration of using other EDD instances for implementation one by one.

**AIGC Expert Integration.** We first consider implementing an AIGC expert to learn the deep generative artifacts. For implementation, we introduce the *SRI model*, based on self-reconstruction images generated by Simswap (Chen et al., 2020) and train on the *Xception model*, designed to capture self-reconstruction generative features. However, from Fig. 8, integrating this model into our framework results in only a minor performance improvement of 0.3%. Further analysis reveals a negative correlation between the model's features and fake labels in the training set (do NOT meet the **Criteria-1**), indicating that these artifacts are poorly represented in the training data. Consequently, the model struggles to leverage the expert-provided features effectively, offering limited benefits over not using the expert model.

**Frequency Expert Integration.** We then integrate a frequency-based model *F3-Net* and train it on the FF++ dataset (Rossler et al., 2019) to capture frequency anomalies. However, from Fig. 8, the overall model's performance is identical to that of the expert, with no improvement. Although the expert features are positively correlated with fake labels, the frequency-based scores are overfitted to the training set and do not accurately reflect the true feature quantity, with only near-1 (1 for fake) and near-0 (0 for real) predictions (do NOT meet the **Criteria-2**) This leads to a *shortcut*, where the model relies solely on the expert's output without learning from the feature information, thus limiting the extendability of the integrated model.

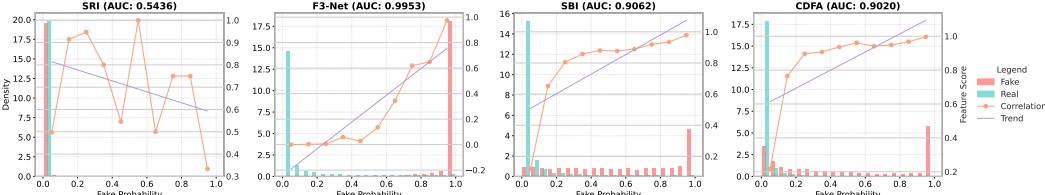

Figure 8: The probability distributions of different expert models on the FF++ training dataset. From left to right, the models are SRI, F3-Net, SBI, and CFDA, corresponding to experts in capturing self-reconstruction, frequency anomalies, and self-blending artifacts, respectively. The blending here directly uses the trained weights.

Table 4: Comparison of methods across datasets with values rounded to two decimal places, where the evaluation metric is AUC. The 'Diff (mllm)' column shows the difference from the mllm average.

| Variant | CDF | DFDCP | DFDC | DFD | Uniface | e4s | Facedancer | FSGAN | Inswap | Simswap | Avg | Diff |
|---------|-----|-------|------|-----|---------|-----|------------|-------|--------|---------|-----|------|
| MLLM | 83.3 | 82.0 | 79.2 | 91.4 | 84.5 | 94.1 | 79.9 | 88.0 | 77.2 | 83.3 | 84.3 | 0.0 |
| SRI | 42.9 | 49.3 | 52.9 | 50.9 | 97.3 | 65.7 | 71.3 | 80.1 | 80.5 | 99.9 | 69.1 | -15.2 |
| SRI+MLLM | 83.2 | 82.5 | 77.6 | 88.8 | 85.6 | **95.8** | 81.9 | 88.5 | 77.9 | 84.6 | 84.6 | +0.3 |
| F3Net | 77.0 | 77.2 | 72.8 | 82.3 | 87.5 | 71.6 | 75.4 | 89.2 | 83.9 | 77.2 | 79.4 | -4.9 |
| F3Net+MLLM | 76.8 | 77.8 | 73.1 | 83.3 | 88.4 | 75.5 | 76.6 | 89.8 | 84.6 | 78.4 | 80.4 | -3.9 |
| SBI | 82.1 | 82.3 | 70.5 | 85.5 | 83.4 | 76.8 | 68.5 | 83.2 | 77.4 | 87.7 | 79.7 | -4.6 |
| SBI+MLLM | 88.6 | 85.5 | 75.6 | 90.8 | **88.7** | 93.7 | 77.6 | 88.2 | 81.6 | **91.1** | 86.2 | +1.9 |
| CDFA | 87.9 | 86.6 | 83.5 | 90.9 | 76.5 | 67.4 | 75.4 | 84.8 | 72.0 | 76.1 | 80.1 | -4.2 |
| CFDA+MLLM | **90.3** | **89.7** | **83.5** | **92.5** | 85.2 | 91.2 | **83.8** | **89.9** | 78.5 | 84.9 | **87.0** | **+2.7** |

**SBI and CFDA Models Integration.** We also integrate another blending-based expert model, *SBI* (Shiohara & Yamasaki, 2022), which specializes in detecting blending artifacts. From Fig. 8, we can see that trained using self-blending techniques on real images to prevent overfitting, the SBI model's

Table 5: Performance Comparison of Different Models on Various Datasets. The **remove 95** and **remove 99.5** scenarios represent extreme cases of data imbalance by removing 95% and 99.5% of the samples near the real distribution, respectively.

| Model | Celeb-DF-v2 | DFDCP | E4S | Facedancer | FSGAN | Inswap | Simswap |
|---|---|---|---|---|---|---|---|
| remove 99.5 | 0.756 | 0.790 | 0.636 | 0.672 | 0.802 | 0.630 | 0.654 |
| remove 95 | 0.793 | 0.814 | 0.689 | 0.697 | 0.821 | 0.657 | 0.703 |
| CDFA+MLLM | **0.903** | **0.896** | **0.912** | **0.838** | **0.899** | **0.785** | **0.849** |

expert features show a strong correlation with fake labels, and its scoring effectively quantifies the extent of blending artifacts. Similarly, the incorporation of the *CFDA model* (Lin et al., 2024), an enhanced version of the SBI model, results in an additional performance boost, indicating that as the expert model's ability to capture blending features improved, the overall model's generalization capability also increases.

To explain **criteria-3**, we conducted additional experiments using non-uniform data distribution. Specifically, we created an extremely imbalanced dataset by removing a large portion of fake samples that do not contain the blending feature. As the imbalance increased, the model's performance degraded, and in extreme cases, it began to rely on shortcut solutions. In the **remove 95** and **remove 99.5** cases, we removed 95% and 99.5% of samples close to the real distribution, respectively, resulting in highly imbalanced datasets with mostly fake samples remaining.

### A.3 Evaluation of Robustness Against Unseen Perturbations

To evaluate the robustness of our model to random perturbations, we follow the methodology outlined in previous studies (Haliassos et al., 2021; Zheng et al., 2021b), which examines four types of degradation: Gaussian blur, block-wise distortion, contrast changes, and JPEG compression. Each perturbation is applied at five different levels to assess the model's performance under varying degrees of distortion.

To highlight the advantages of our approach over conventional detectors like FWA (Li, 2018), SBI (Shiohara & Yamasaki, 2022), and X-ray (Li et al., 2020a), we conducted multiple evaluations. As illustrated in Figure 9, which shows the video-level AUC results for these unseen perturbations using a model trained on FF++ c23, our method consistently demonstrates superior robustness compared to other RGB-based methods.

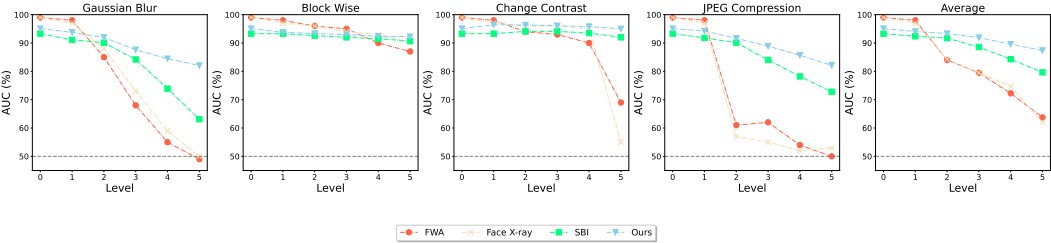

Figure 9: Robustness evaluation. We adopt four types of degradation for examining the robustness of our model: Gaussian blur, block-wise distortion, contrast changes, and JPEG compression. Our model shows superior robustness over other compared models.

### A.4 Human Study

We begin by randomly selecting a balanced proportion of images from the current deepfake dataset before the evaluation starts. The evaluation will be conducted from three aspects: **detection**, **explanation**, and **detection with explanation**.

**Detection.** In this aspect, we use the prompts provided in the paper (**?**) to guide the model to make a decision regarding whether the image is real or fake. This section is handled by the experiment designers.

**Explanation Performance for Human Preference.** For the evaluation based on human preference, well-selected images will be provided along with their ground truth. The model is tasked with generating corresponding explanations. Participants will then choose between the answers from three different models.

**Detection and Explanation Performance Evaluation.** In this evaluation, the ground truth of the image is provided to the participants, but the model is unaware of it. After receiving an image, the model is required to give both a **detection** (whether the image is real or fake) and an **explanation**.

For GPT-4o's evaluation of detection and preference-based assessment of explanation, as well as combined detection and explanation, the prompt is illustrated in Fig 13.

**Supplementary details on human study.** We select *well-educated participants* and provide them with *detailed guidelines* on deepfake technology prior to the experiment to ensure the reliability of the human study results. Furthermore, potential risks associated with the experiment are carefully evaluated, and approval is obtained through the relevant ethical review process. Fig. 10, Fig. 11, and Fig. 12 include details of the experimental setup and the ethical review process, ensuring the reliability of our study.

```
Project Summary:

With the rapid development of Deepfake technology, the generation of fake
images and videos has become increasingly realistic, posing significant
threats to society and individual privacy. Despite the availability of
various deep learning models for detecting Deepfake content, there are still
notable shortcomings in terms of explainability and transparency. These
differences directly impact human trust and understanding during the
detection process.

This project aims to evaluate the alignment between our model's
```

```
explainability in Deepfake detection and human intuition through a human
evaluation study. We will compare our model with current mainstream Deepfake
detection models to examine the intuitiveness, accuracy, and user trust in
the explanations provided. The study will involve participants evaluating
the detection results and explanations from different models, assessing the
effectiveness of each model in real-world applications.
```

Figure 10: Human study material part1

A.5 EXPERIMENTS ON DIFFERENT LLMS/MLLMS

We conducted experiments using various models, including GPT4o (Achiam et al., 2023), Phi-3-vision (Abdin et al., 2024), Claude3.5-Sonnet (Abdin et al., 2024), and LLaVa (Liu et al., 2024), to evaluate the adaptability and robustness of our framework.

Table 6: Experiments on different LLMs/MLLMs were conducted to evaluate their performance under various conditions. The evaluation metric used for these experiments is the Area Under the Curve (AUC)

| Variant | CDF | DFDCP | DFDC | DFD | Uniface | e4s | Facedancer | FSGAN | Inswap | Simswap | Avg |
|---|---|---|---|---|---|---|---|---|---|---|---|
| GPT4o + Phi-3-vision | 88.6 | 87.1 | 83.5 | 90.9 | 81.8 | 77.5 | 78.8 | 85.7 | 77.5 | 80.6 | 83.2 |
| GPT4o + LLaVa-7B | 90.3 | 89.7 | 83.5 | 92.5 | 85.2 | 91.2 | 83.8 | 89.9 | 78.5 | 84.9 | 87.0 |
| Claude3.5-Sonnet + LLaVa-7B | 88.8 | 88.5 | 82.6 | 92.7 | 84.6 | 90.1 | 83.8 | 89.7 | 79.5 | 85.6 | 86.6 |
| GPT4o + LLaVa-13B | 91.3 | 90.3 | 83.4 | 92.5 | 86.0 | 92.5 | 84.5 | 91.0 | 80.6 | 85.4 | 87.8 |

**Different LLMs to generate questions in FMA.** In the FMA stage, we employed different LLMs, such as GPT4o and Claude 3.5-Sonnet, to generate forgery-related questions and test the adaptability

```
Project Significance:

This research will reveal the strengths and weaknesses of different Deepfake
detection models in terms of explainability, especially in alignment with
human intuition. By comparing our model with other mainstream models, we
hope to demonstrate that our model is not only competitive in detection
accuracy but also achieves higher trust and satisfaction in terms of
explainability. The project outcomes will offer new insights and directions
for the future design and application of Deepfake detection models,
promoting the development of more transparent and reliable detection
technologies.

Potential Side Effects, Hazards, and Emergency Plans:

Side Effects: During the project, participants may experience confusion
or reduced trust in real images due to exposure to a large amount of Deepfake
content, leading to difficulty in distinguishing between real and fake
images.
Emergency Plan: We will inform participants that there are currently
various effective Deepfake detection solutions, including the model we are
developing, which can effectively counter Deepfake attacks to some extent.
Through education and explanation, we will help participants understand the
progress of the technology and the available protective measures, enhancing
their trust in real images.
```

Figure 11: Human study material part2

of our framework. The results, shown in *GPT4o + LLaVa-7B* and *Claude 3.5-Sonnet + LLaVa-7B*, demonstrate consistent performance regardless of the LLM used. Questions from Claude 3.5-Sonnet were also effective (see Tabs. 15 and 16).

**Different sizes of MLLMs for fine-funing in SFS and WFS.** In the SFS and WFS stages, we investigate the impact of using different sizes of MLLMs with the same architecture during fine-tuning. For instance, we compare *GPT4o + LLaVa-7B* and *GPT4o + LLaVa-13B*. The results indicate that as the size of the model increases, the performance of the framework improves proportionally, benefiting from the enhanced capabilities of the larger MLLMs.

**Different MLLMs for fine-tuning in SFS and WFS.** We also examine the effect of using different MLLMs during the SFS and WFS fine-tuning stages to determine whether our framework relies on a specific MLLM. For example, we compare *GPT4o + Phi-3-vision* and *GPT4o + LLaVa-13B*. The results demonstrate that our framework is not dependent on a specific MLLM and that different MLLMs can benefit from it.

**Summary of findings.** These experiments collectively highlight the robustness of our framework. It is not dependent on specific LLMs or MLLMs, making it adaptable to a wide range of models. Furthermore, as the performance and size of the underlying models improve, our framework effectively leverages these advancements to achieve enhanced results.

### A.6    IN-DOMAIN RESULTS IN THE FF++ DATASET

In our manuscript, we mainly focus on the cross-domain evaluation to assess the generalization performance of different models. Here, we conduct the in-domain evaluation on the FF++ dataset and compare our approach with the other four SOTA methods: FWA, Face X-ray, SRM, and CDFA. Following DeepfakeBench Yan et al. (2023d), we train all models on FF++ (c23) and test them on FF++ (c23), FF++ (c40), FF-DF, FF-F2F, FF-FS, and FF-NT. As shown in Table 7, the in-domain results demonstrate that our framework achieves the best performance, outperforming all other methods.

```
Potential Ethical Issues and Countermeasures (including Informed Consent,
Privacy Protection, Physical Harm, and Benefit Distribution):

Informed Consent: Participants need to fully understand the research
content, purpose, and potential impact. We will ensure that all participants
voluntarily sign informed consent forms. Countermeasure: Provide detailed
research explanations and Q&A sessions to ensure participants fully
understand and agree to participate.

Privacy Protection: All participant information and data involved in the
research will be kept strictly confidential and will not be used for any
purposes other than the research. Countermeasure: Implement data
encryption and anonymization to ensure participant privacy is not
compromised.

Physical Harm: Although this study mainly involves psychological and
cognitive assessments, we will minimize potential psychological impacts on
participants and avoid any form of psychological stress or harm.
Countermeasure: Conduct real-time monitoring during the study to ensure
participant mental health, and provide participants with the right to
withdraw from the study at any time.

Benefit Distribution: Ensure that participants are not unfairly treated
during the study and that the outcomes of the research do not
disproportionately benefit or disadvantage any individual or group.
Countermeasure: Fair and reasonable distribution of research outcomes,
ensuring openness and transparency. The research results will primarily
focus on academic and social contributions, not for personal or commercial
gain.
```

Figure 12: Human study material part3

Table 7: In-domain results in the FF++ dataset (AUC)

| Detector | FF++$c$23 | FF++$c$40 | FF-DF | FF-F2F | FF-FS | FF-NT | AVG |
|---|---|---|---|---|---|---|---|
| FWA (Li, 2018) | 87.7 | 73.6 | 92.1 | 90.0 | 88.4 | 81.2 | 85.5 |
| Face X-ray (Li et al., 2020a) | 95.9 | 79.3 | 97.9 | 98.7 | 98.7 | 92.9 | 93.9 |
| SRM (Luo et al., 2021b) | 95.8 | 81.1 | 97.3 | 97.0 | 97.4 | 93.0 | 93.6 |
| CDFA (Lin et al., 2024) | 90.2 | 69.0 | 99.9 | 86.9 | 93.3 | 80.7 | 90.2 |
| ours | 96.6 | 82.6 | 99.9 | 97.2 | 98.1 | 91.0 | 94.2 |

## A.7 FINDING IN MLLMS

**The capabilities of large models go far beyond this.** Based on our experimental results, we found that training for multiple epochs can continuously improve performance. However, in our experiments, training for just one epoch already achieved the desired results. Therefore, the experiments presented in the main table are based solely on the results from training for one epoch. Since this performance improvement is not part of the method we proposed, I believe it should not be included in the main table.

Table 8: Model Performance with Varying Epochs, where the evaluation metric is AUC

| Variant | CDF | DFDCP | DFDC | DFD | Uniface | e4s | Facedancer | FSGAN | Inswap | Simswap | Avg |
|---|---|---|---|---|---|---|---|---|---|---|---|
| one epoch | 90.3 | 89.7 | 83.5 | **92.5** | 84.9 | 91.2 | 83.5 | 89.9 | 78.5 | 84.9 | 87.0 |
| two epochs | 92.7 | 89.3 | 83.9 | 91.5 | 87.4 | 93.0 | 84.6 | 89.9 | 81.0 | 86.1 | 88.0 |
| three epochs | **92.7** | **90.9** | **84.5** | 90.4 | **87.8** | **93.6** | **85.9** | **90.0** | **81.1** | **86.6** | **88.4** |

**Effect of inconsistent use of supplementary features in training and inference.** The model performs best when supplementary features are used consistently during both training and inference (average score: **0.8797**), indicating that these features significantly enhance performance. When supplementary features are omitted entirely from both stages, the performance drops (average score: **0.8328**), though it remains better than when features are used inconsistently. Specifically, when features are used during training but not inference, the performance suffers greatly (average score: **0.7661**), suggesting the model relies on these features and struggles without them at inference time. On the other hand, when features are introduced at inference but not used during training, the model achieves slightly better results (average score: **0.8247**), but it cannot fully leverage unseen features, showing the importance of using supplementary features consistently across both phases.

Table 9: Impact of Omitting Supplementary Features During Training and Adding Them During Inference, on Model Performance

|  | Celeb-DF-v2 | DFD | DFDC | DFDCP | DFR | WDF | FFIW | Avg |
|---|---|---|---|---|---|---|---|---|
| no train + no infer | 0.8324 | 0.9140 | 0.7922 | 0.8197 | 0.9371 | 0.7682 | 0.7663 | 0.8328 |
| train + no infer | 0.7649 | 0.8469 | 0.7102 | 0.7203 | 0.8950 | 0.7130 | 0.7127 | 0.7661 |
| no train + infer | 0.8171 | 0.9062 | 0.7906 | 0.8134 | 0.9262 | 0.7472 | 0.7733 | 0.8247 |
| train + infer | **0.9062** | **0.9232** | **0.8300** | **0.8873** | **0.9761** | **0.8144** | **0.8167** | **0.8797** |

**Extension in new datasets.** Our model, trained on a mixture of datasets including FF++, showed improved overall performance when we added a new dataset without blending artifacts to the training process. This demonstrates that incorporating diverse datasets with supplementary features, even from different domains, enhances the model's generalization and comprehensive performance.

Table 10: Comparison of Model Performance When Trained on FF++ Alone vs. FF++ and SRI Across Different Datasets

| Variant | CDF | DFDCP | DFDC | DFD | Uniface | e4s | Facedancer | FSGAN | Inswap | Simswap |
|---|---|---|---|---|---|---|---|---|---|---|
| Train FF++ | 90.3 | **89.7** | 83.5 | 92.4 | 85.2 | 91.2 | 83.8 | 89.9 | 78.4 | 84.9 |
| Train FF++ and SRI | **91.5** | 89.3 | **83.9** | **92.7** | **87.4** | **93.0** | **84.6** | **89.9** | **81.0** | **86.1** |

## A.8 IMPLEMENTATION DETAILS

We use the LLava model and a 4090 GPU, with image cropping following the method from DeepfakeBench. AUC is calculated by directly obtaining the token probabilities. Previous AUC calculations for large models mostly relied on averaging methods, such as in (**?**), but this approach is not very accurate because: (1) multiple samplings are needed to approximate the true probability distribution, and (2) large models inherently perform inference with a default temperature, which itself involves sampling over probabilities. Averaging over multiple samples effectively results in a second layer of sampling, making the evaluation less accurate. Therefore, in this paper, we calculate AUC by directly obtaining token probabilities.

## A.9 SAMPLE SHOWING

Here, I will present some failure cases of Pre-trained MLLMs, followed by a comparison with our results.

For LLaVa, we use the same prompts as GPT-4 to ensure fairness in the evaluation process. The robustness of Llava in these tasks is illustrated in Figure 14.

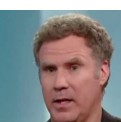

Label: real

Detection

**USER:** I want you to work as an image forensic expert for Deepfake faces. Check if the image has the artifact attribute listed in the following list and ONLY return the attribute number in this image. The artifact list is [1-asymmetric eye iris; 2-irregular glasses shape or reflection; 3-irregular teeth shape or texture; 4-irregular ears or earrings; 5-strange hair texture; 6-inconsistent skin texture; 7-inconsistent lighting and shading; 8-strange background; 9-weird hands; 10-unnatural edges].

**GPT: Attribute number**: Based on the provided list of artifacts, I reviewed the image and found that it displays the following attribute:**5-strange hair texture.** The texture of the hair appears somewhat artificial, a common characteristic of AI-generated images.

**USER:** This image is real, tell me the reason.

Explanation

**GPT:** The image you provided appears to be a real person, possibly from a television interview or public appearance. The reason it seems real is due to the natural lighting, human facial expressions, and other small details such as skin texture and natural positioning, which are characteristics of an authentic photograph of a person in a live setting. However, based on this image alone, I cannot confirm its exact source or context.

**USER:** Is this image is real or deepfake, tell me the reason.

Detection&
Explanation

**GPT:** The image you provided appears to be a real person, possibly from a television interview or public appearance. The reason it seems real is due to the natural lighting, human facial expressions, and other small details such as skin texture and natural positioning, which are characteristics of an authentic photograph of a person in a live setting. However, based on this image alone, I cannot confirm its exact source or context.

Figure 13: Diagram showing how the GPT4o generates results for **detection**, **explanation**, and **detection with explanation**.

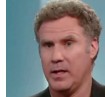

Label: real

Detection

**USER:** I want you to work as an image forensic expert for Deepfake faces. Check if the image has the artifact attribute listed in the following list and ONLY return the attribute number in this image. The artifact list is [1-asymmetric eye iris; 2-irregular glasses shape or reflection; 3-irregular teeth shape or texture; 4-irregular ears or earrings; 5-strange hair texture; 6-inconsistent skin texture; 7-inconsistent lighting and shading; 8-strange background; 9-weird hands; 10-unnatural edges].

**LLaVa: 1**

**USER:** This image is real, tell me the reason.

Explanation

**LLaVa:** The reason for the image being real is related to the subject matter depicted in the image, which is a man with a beard and mustache. The man appears to be the famous actor, John Wayne, known for his distinctive facial appearance in movies. The image is a close-up, and John Wayne's facial features are clearly visible and recognizable. It's not a photo-shopped or digitally altered image. The context of the image is John Wayne talking to someone, which is also consistent with his well-known roles.

**USER:** Is this image is real or deepfake, tell me the reason.

Detection&
Explanation

**LLaVa:** The image is real, as it is an authentic photo of John Wayne, a famous actor known for his distinctive beard and mustache. The close-up view of John Wayne's face in the image allows for clear recognition of his facial features, making it a genuine representation of him.

Figure 14: Diagram showing how the LLaVa generates results for **detection**, **explanation**, and **detection with explanation**.

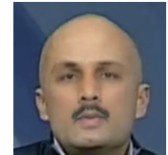

Label: real

Detection

**USER:** Is this image real or fake?

**OURS:** This image is real. The image appears to be real, as the man's facial features, such as his nose, eyes, and mouth, are well-defined and not distorted. The skin tone appears natural, and there are no signs of blurriness or unusual lighting. The man's facial expressions and overall appearance seem genuine, making it likely that the image is a true representation of the man. And this image does not contain any blending artifacts.

**USER:** This image is real, tell me the reason.

Explanation

**OURS:** The image of the man's face appears to be real, as it displays a well-defined facial structure with clear skin, prominent eyes, and a nose that is neither overly smooth nor has flat cheekbones. The man's facial features are well-defined, and there is no noticeable blurriness or distortion in the image. The overall appearance of the man's face suggests that it is a genuine representation of the individual. And this image does not contain any blending artifacts.

**USER:** Is this image real or fake?

Detection& Explanation

**OURS:** This image is real. The image appears to be real, as the man's facial features, such as his nose, eyes, and mouth, are well-defined and not distorted. The skin tone appears natural, and there are no signs of blurriness or unusual lighting. The man's facial expressions and overall appearance seem genuine, making it likely that the image is a true representation of the man. And this image does not contain any blending artifacts.

Figure 15: Diagram showing how the ours model generates results for **detection**, **explanation**, and **detection with explanation**.

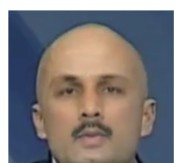

Fake

**USER:** Is this image real or fake? Tell the reason in detail?

**MLLM:** The image is real, featuring a bald man wearing a blue shirt and a white collar. He is talking and has a mustache. The image is not photoshopped. The man appears to be a news reporter or a presenter, and his expression suggests that he is engaged in a discussion or interview. The image is a genuine representation of the man and his attire.

Fake

**USER:** This is fake image, tell the reason in detail.

**MLLM:** The image is a fake or manipulated picture of a bald man with a goatee, likely created for entertainment or artistic purposes. The reason for this manipulation could be to create a humorous or unconventional visual effect, to make a statement, or to generate curiosity and discussion. The image may have been created using photoshop software, resulting in the unusual combination of a bald man with a goatee. This manipulation goes beyond the typical representation of a bald man, as it adds a distinctive element to the image that is not commonly seen in real life.

Figure 16: Bad sample of Pre-trained model (part1)

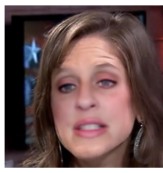

**USER:** Is this image real or fake? Tell the reason in detail.

**MLLM:** The image is real, as it shows a woman with blue eyes and red lips. The image also features a TV in the background, which is an indication that the setting is likely a TV studio or a live broadcast. The presence of the TV and the woman's makeup suggest that this is a real, live situation, rather than a fake or manipulated image.

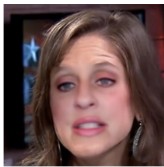

Fake

**USER:** Why this image is fake?

**MLLM:** The image is fake because the person's face appears to be photoshopped, This can be inferred from the fact that her eyes do not appear to be looking at the camera, and the lighting and background are not consistent. Additionally, the woman's face is distorted, and she has fake eyelashes and a blurry face. The face manipulation creates an unrealistic and artificial appearance, making it clear that the image is not a genuine photograph.

Figure 17: Bad sample of Pre-trained model (part2)

Table 11: Relationship between various facial features and deepfake detection (Part 1)

| No. | Question | Pretrain |
|---|---|---|
| 1 | Is the face color related to deepfake detection? | No |
| 2 | Are the eyes related to deepfake detection? | No |
| 3 | Are the facial features related to deepfake detection? | No |
| 4 | Is the nose contour related to deepfake detection? | No |
| 5 | Is face blurriness related to deepfake detection? | No |
| 6 | Is the skin tone related to deepfake detection? | No |
| 7 | Are the cheeks related to deepfake detection? | No |
| 8 | Is the skin tone pattern related to deepfake detection? | No |
| 9 | Is the placement of facial features related to deepfake detection? | No |
| 10 | Are the lips related to deepfake detection? | Yes |
| 11 | Is facial symmetry related to deepfake detection? | No |
| 12 | Is the lighting on the cheeks related to deepfake detection? | Yes |
| 13 | Is the facial lighting related to deepfake detection? | No |
| 14 | Are the shapes of facial features related to deepfake detection? | No |
| 15 | Is facial evenness related to deepfake detection? | Yes |
| 16 | Are the cheekbones related to deepfake detection? | No |
| 17 | Is the face layout related to deepfake detection? | No |
| 18 | Are the lip edges related to deepfake detection? | No |
| 19 | Is facial detail related to deepfake detection? | Yes |
| 20 | Is cheek smoothness related to deepfake detection? | No |
| 21 | Is the forehead shape related to deepfake detection? | No |
| 22 | Is face-background blending related to deepfake detection? | Yes |
| 23 | Is skin texture related to deepfake detection? | No |
| 24 | Are the eyelashes related to deepfake detection? | No |
| 25 | Are facial lines related to deepfake detection? | No |
| 26 | Is facial expression related to deepfake detection? | No |
| 27 | Is the nose shape related to deepfake detection? | No |
| 28 | Are color changes on the face related to deepfake detection? | No |
| 29 | Is the mouth shape related to deepfake detection? | No |
| 30 | Are the face edges related to deepfake detection? | No |
| 31 | Is facial rigidity related to deepfake detection? | No |
| 32 | Are sharp facial lines related to deepfake detection? | No |
| 33 | Is skin perfection related to deepfake detection? | No |
| 34 | Is forehead shininess related to deepfake detection? | Yes |
| 35 | Are sharp face edges related to deepfake detection? | Yes |
| 36 | Is skin smoothness related to deepfake detection? | No |
| 37 | Are eye details related to deepfake detection? | No |
| 38 | Are smooth facial lines related to deepfake detection? | No |
| 39 | Is lip texture related to deepfake detection? | Yes |
| 40 | Is forehead shine evenness related to deepfake detection? | No |
| 41 | Are the eyebrows related to deepfake detection? | No |
| 42 | Are unusual eye appearances related to deepfake detection? | No |
| 43 | Are facial transitions related to deepfake detection? | Yes |
| 44 | Is face color related to deepfake detection? | No |
| 45 | Is facial emotion exaggeration related to deepfake detection? | No |
| 46 | Is unusual face layout related to deepfake detection? | No |
| 47 | Are eye reflections related to deepfake detection? | No |
| 48 | Is skin texture roughness related to deepfake detection? | No |
| 49 | Is the jawline related to deepfake detection? | No |
| 50 | Is facial expression stiffness related to deepfake detection? | Yes |

Table 12: Relationship between various facial features and deepfake detection (Part 2)

| No. | Question | Pretrain |
|-----|----------|----------|
| 51 | Is nose texture related to deepfake detection? | No |
| 52 | Is skin shininess under the nose related to deepfake detection? | No |
| 53 | Is uneven facial sharpness related to deepfake detection? | Yes |
| 54 | Is facial blending related to deepfake detection? | No |
| 55 | Is facial lighting evenness related to deepfake detection? | No |
| 56 | Is nose bridge smoothness related to deepfake detection? | No |
| 57 | Is the hairline related to deepfake detection? | No |
| 58 | Is skin texture evenness related to deepfake detection? | No |
| 59 | Is facial feature balance related to deepfake detection? | No |
| 60 | Is facial symmetry related to deepfake detection? | No |
| 61 | Is forced facial expression related to deepfake detection? | No |
| 62 | Are the nostrils related to deepfake detection? | No |
| 63 | Are unnatural lip appearances related to deepfake detection? | No |
| 64 | Is partial skin smoothness related to deepfake detection? | No |
| 65 | Is lip texture related to deepfake detection? | No |
| 66 | Is lighting around the nose related to deepfake detection? | Yes |
| 67 | Are facial feature proportions related to deepfake detection? | Yes |
| 68 | Is skin smoothness around the nose related to deepfake detection? | No |
| 69 | Are soft facial creases related to deepfake detection? | No |
| 70 | Are teeth appearances related to deepfake detection? | No |
| 71 | Is neck-face transition related to deepfake detection? | No |
| 72 | Is skin tone variation related to deepfake detection? | No |
| 73 | Is face edge sharpness related to deepfake detection? | No |
| 74 | Is chin outline visibility related to deepfake detection? | No |
| 75 | Is facial lighting evenness related to deepfake detection? | Yes |
| 76 | Are ear details related to deepfake detection? | No |
| 77 | Is chin smoothness related to deepfake detection? | No |
| 78 | Are bright facial areas related to deepfake detection? | No |
| 79 | Is skin brightness near the mouth related to deepfake detection? | No |
| 80 | Are nostril appearances related to deepfake detection? | No |
| 81 | Are dimples related to deepfake detection? | Yes |
| 82 | Is jawline prominence related to deepfake detection? | No |
| 83 | Is under-eye texture related to deepfake detection? | No |
| 84 | Is facial blending related to deepfake detection? | Yes |
| 85 | Is chin shadow related to deepfake detection? | No |
| 86 | Are forehead shadows related to deepfake detection? | No |
| 87 | Is nose light reflection related to deepfake detection? | No |
| 88 | Is face-background transition related to deepfake detection? | No |
| 89 | Is forehead light reflection related to deepfake detection? | No |
| 90 | Are nose shadows related to deepfake detection? | No |
| 91 | Is lighting around the mouth related to deepfake detection? | No |
| 92 | Is neck smoothness related to deepfake detection? | No |
| 93 | Are face outlines related to deepfake detection? | No |
| 94 | Are face edges related to deepfake detection? | No |
| 95 | Are skin details related to deepfake detection? | No |
| 96 | Are under-eye shadows related to deepfake detection? | No |
| 97 | Are cheek shadows related to deepfake detection? | No |
| 98 | Are cheekbone appearances related to deepfake detection? | No |
| 99 | Is facial lighting related to deepfake detection? | No |
| 100 | Are facial wrinkle details related to deepfake detection? | No |

Table 13: Top Strong 50 Features

| Rank | Question | Pretrain | Strengthened |
|------|----------|----------|--------------|
| 1 | Is the face color unusual? | 0.6340 | 0.7486 |
| 2 | Is there something wrong with the eyes? | 0.6309 | 0.6320 |
| 3 | Do the facial features look oddly shaped? | 0.6292 | 0.6636 |
| 4 | Is the contour of the nose incorrect? | 0.6278 | 0.5817 |
| 5 | Is part of the face blurry? | 0.6231 | 0.7643 |
| 6 | Does the skin tone make the face look fake? | 0.6165 | 0.6479 |
| 7 | Is there something wrong with the cheek? | 0.6144 | 0.8082 |
| 8 | Are there strange patterns in the skin tone? | 0.6144 | 0.8075 |
| 9 | Are the face parts out of place? | 0.6130 | 0.7919 |
| 10 | Do the lips seem out of place or strangely shaped? | 0.6127 | 0.7408 |
| 11 | Is one side of the face uneven with the other? | 0.6123 | 0.7622 |
| 12 | Are there strange lighting spots on the cheeks? | 0.6111 | 0.8029 |
| 13 | Does the lighting change strangely on the face? | 0.6092 | 0.8006 |
| 14 | Are the shapes of the eyes, nose, or mouth unnatural? | 0.6054 | 0.6714 |
| 15 | Does the face look uneven or off? | 0.6048 | 0.6732 |
| 16 | Does the cheekbone appear too flat? | 0.6014 | 0.7406 |
| 17 | Does the face layout look wrong? | 0.5986 | 0.5422 |
| 18 | Are the edges of the lips too smooth? | 0.5979 | 0.6264 |
| 19 | Is part of the face lacking detail? | 0.5942 | 0.6843 |
| 20 | Are the cheeks too smooth? | 0.5934 | 0.6728 |
| 21 | Does the forehead look odd in shape? | 0.5911 | 0.7493 |
| 22 | Does the face mix poorly with the background? | 0.5902 | 0.6382 |
| 23 | Is the skin texture uneven? | 0.5861 | 0.7158 |
| 24 | Are the eyelashes missing or blurred? | 0.5857 | 0.6733 |
| 25 | Are the face lines uneven or changing in different areas? | 0.5826 | 0.6546 |
| 26 | Does the face lack expression? | 0.5822 | 0.6679 |
| 27 | Does the nose shape look odd? | 0.5812 | 0.5306 |
| 28 | Are the color changes on the face and skin sudden? | 0.5807 | 0.6643 |
| 29 | Does the mouth appear too flat? | 0.5775 | 0.6542 |
| 30 | Are the edges of the face too sharp? | 0.5774 | 0.8188 |
| 31 | Does the face appear too rigid? | 0.5770 | 0.7446 |
| 32 | Are the face lines too sharp? | 0.5761 | 0.7724 |
| 33 | Does the skin look too perfect, like it was edited? | 0.5755 | 0.5749 |
| 34 | Is the forehead too shiny? | 0.5737 | 0.8168 |
| 35 | Are the face edges too sharp? | 0.5720 | 0.8162 |
| 36 | Does the face skin look too smooth? | 0.5640 | 0.5306 |
| 37 | Are the eyes blurry or lacking detail? | 0.5636 | 0.5362 |
| 38 | Are the face lines too smooth? | 0.5549 | 0.5927 |
| 39 | Are the lips too smooth or lacking texture? | 0.5537 | 0.5475 |
| 40 | Is the forehead's shine uneven? | 0.5515 | 0.7208 |
| 41 | Are the eyebrows too dark or too light? | 0.5454 | 0.5075 |
| 42 | Do the eyes look odd? | 0.5433 | 0.5389 |
| 43 | Are transitions on the face poorly blended? | 0.5410 | 0.5854 |
| 44 | Do the face colors look strange? | 0.5382 | 0.6163 |
| 45 | Does the face show emotions that seem exaggerated? | 0.5355 | 0.6337 |
| 46 | Does the face layout look unusual? | 0.5344 | 0.5377 |
| 47 | Do the eyes have unnatural reflections? | 0.5323 | 0.6417 |
| 48 | Does the face have rough or uneven skin texture? | 0.5292 | 0.7973 |
| 49 | Does the jawline appear too sharp or unclear? | 0.5292 | 0.5017 |
| 50 | Does the facial expression look stiff? | 0.5289 | 0.5346 |

Table 14: Bottom 50 Weak Features

| Rank | Question | Pretrained | Strengthened |
|---|---|---|---|
| 51 | Does the nose lack texture? | 0.5231 | 0.5111 |
| 52 | Is the skin too shiny under the nose? | 0.5223 | 0.7458 |
| 53 | Is the sharpness of the face uneven in parts? | 0.5214 | 0.5701 |
| 54 | Does the blending on the face look unnatural or uneven? | 0.5212 | 0.5151 |
| 55 | Is the lighting on the face strange or uneven? | 0.5200 | 0.5968 |
| 56 | Does the nose bridge appear too smooth? | 0.5172 | 0.5395 |
| 57 | Does the hairline seem unnatural? | 0.5148 | 0.5495 |
| 58 | Does the face skin texture look uneven? | 0.5144 | 0.5489 |
| 59 | Do the face parts look out of balance? | 0.5137 | 0.5887 |
| 60 | Are the facial features too symmetrical? | 0.5130 | 0.7300 |
| 61 | Does the facial expression look forced? | 0.5116 | 0.5562 |
| 62 | Are the nostrils hard to see? | 0.5115 | 0.6535 |
| 63 | Do the lips look unnatural? | 0.5110 | 0.5555 |
| 64 | Does the face skin look too smooth in some areas? | 0.5089 | 0.5287 |
| 65 | Do the lips lack natural texture? | 0.5083 | 0.5855 |
| 66 | Is the lighting around the nose inconsistent? | 0.5080 | 0.7257 |
| 67 | Do the sizes of the eyes, nose, and mouth seem off? | 0.5038 | 0.5275 |
| 68 | Does the skin around the nose look unnaturally smooth? | 0.5030 | 0.5309 |
| 69 | Are the facial creases too soft? | 0.5028 | 0.7943 |
| 70 | Do the teeth appear blurry or unrealistic? | 0.5028 | 0.5210 |
| 71 | Is the transition between the neck and the face not smooth? | 0.5026 | 0.5330 |
| 72 | Is the skin tone different in parts of the face? | 0.5023 | 0.5749 |
| 73 | Does the face lack sharpness around the edges? | 0.5021 | 0.5311 |
| 74 | Is the chin outline hard to see? | 0.5021 | 0.6160 |
| 75 | Is the lighting uneven on the face? | 0.5012 | 0.6351 |
| 76 | Are the details around the ears unclear? | 0.5010 | 0.6751 |
| 77 | Is the chin too smooth compared to the rest of the face? | 0.5010 | 0.5664 |
| 78 | Do the bright areas on the face seem odd? | 0.5007 | 0.5196 |
| 79 | Is the skin near the mouth unnaturally bright? | 0.5007 | 0.5930 |
| 80 | Are the nostrils blurry or unclear? | 0.5007 | 0.5125 |
| 81 | Are the dimples missing or poorly defined? | 0.5005 | 0.5000 |
| 82 | Is the jawline too pronounced or too faint? | 0.5000 | 0.5003 |
| 83 | Is the area under the eyes missing natural texture? | 0.5000 | 0.5111 |
| 84 | Is there blending on the face that looks edited? | 0.5000 | 0.5014 |
| 85 | Does the shadow under the chin seem unnatural? | 0.5000 | 0.5090 |
| 86 | Is the forehead missing natural shadows? | 0.5000 | 0.5000 |
| 87 | Does the light reflection on the nose look strange? | 0.5000 | 0.5049 |
| 88 | Are the transitions between the face and the background poorly blended? | 0.5000 | 0.5447 |
| 89 | Does the light reflection on the forehead look artificial? | 0.5000 | 0.5007 |
| 90 | Are there missing shadows around the nose? | 0.5000 | 0.5217 |
| 91 | Does the lighting around the mouth look unusual? | 0.5000 | 0.5301 |
| 92 | Does the neck look unnaturally smooth compared to the face? | 0.5000 | 0.6259 |
| 93 | Do the face outlines look off? | 0.5000 | 0.5247 |
| 94 | Do the edges around the face look unnatural? | 0.5000 | 0.5299 |
| 95 | Are the fine details on the skin missing? | 0.5000 | 0.5165 |
| 96 | Are the shadows under the eyes missing? | 0.5000 | 0.5000 |
| 97 | Are the cheeks lacking shadows? | 0.5000 | 0.5000 |
| 98 | Do the cheekbones appear unnaturally smooth? | 0.5000 | 0.5709 |
| 99 | Does the face appear overly lit in certain areas? | 0.5000 | 0.6758 |
| 100 | Are the wrinkles on the face lacking detail? | 0.5000 | 0.5014 |

Table 15: Questions list generated by Claude3.5-Sonnet (part1)

| No. | Question |
|-----|----------|
| 1 | Are there noticeable inconsistencies in facial symmetry? Return me yes or no |
| 2 | Does the skin texture appear artificially smooth or lacking natural details? Return me yes or no |
| 3 | Are the eyes misaligned or disproportionate? Return me yes or no |
| 4 | Is there unnatural blending between facial features and background? Return me yes or no |
| 5 | Do shadows and lighting appear inconsistent across the face? Return me yes or no |
| 6 | Are facial expressions unnatural or mechanically rigid? Return me yes or no |
| 7 | Does the hairline show signs of artificial blending? Return me yes or no |
| 8 | Are there visible artifacts or glitches in the image? Return me yes or no |
| 9 | Do reflections in the eyes match the environment? Return me yes or no |
| 10 | Is there proper alignment of facial features? Return me yes or no |
| 11 | Does the skin show natural imperfections and pores? Return me yes or no |
| 12 | Are teeth shapes and alignment realistic? Return me yes or no |
| 13 | Is there consistent image quality across all facial areas? Return me yes or no |
| 14 | Do facial proportions follow natural human anatomy? Return me yes or no |
| 15 | Are shadows cast appropriately based on lighting? Return me yes or no |
| 16 | Does facial hair follow natural growth patterns? Return me yes or no |
| 17 | Is there proper depth and dimension to facial features? Return me yes or no |
| 18 | Are color tones consistent throughout the face? Return me yes or no |
| 19 | Do glasses and accessories appear properly attached? Return me yes or no |
| 20 | Is there natural variation in skin texture? Return me yes or no |
| 21 | Are facial contours anatomically correct? Return me yes or no |
| 22 | Does the head size match body proportions? Return me yes or no |
| 23 | Is there appropriate detail in fine features? Return me yes or no |
| 24 | Are transitions between features naturally blended? Return me yes or no |
| 25 | Do facial movements appear fluid and natural? Return me yes or no |
| 26 | Are ear shapes and positions symmetrical? Return me yes or no |
| 27 | Do eyebrows have natural hair patterns? Return me yes or no |
| 28 | Is there consistent resolution between face and background? Return me yes or no |
| 29 | Are nose contours anatomically accurate? Return me yes or no |
| 30 | Does makeup application appear natural? Return me yes or no |
| 31 | Are facial wrinkles and lines age-appropriate? Return me yes or no |
| 32 | Do eyelashes appear realistic and properly attached? Return me yes or no |
| 33 | Is there natural skin coloration variation? Return me yes or no |
| 34 | Are facial highlights consistent with lighting? Return me yes or no |
| 35 | Do lips have natural texture and color? Return me yes or no |
| 36 | Is there proper depth in eye sockets? Return me yes or no |
| 37 | Are facial moles and marks naturally placed? Return me yes or no |
| 38 | Do teeth have individual characteristics? Return me yes or no |
| 39 | Is there natural asymmetry in facial features? Return me yes or no |
| 40 | Are skin pores visible where expected? Return me yes or no |
| 41 | Do facial muscles move naturally? Return me yes or no |
| 42 | Is there consistent focus across the image? Return me yes or no |
| 43 | Are shadows under facial features natural? Return me yes or no |
| 44 | Do earrings and jewelry sit naturally? Return me yes or no |
| 45 | Is there proper skin subsurface scattering? Return me yes or no |
| 46 | Are facial proportions consistent in different angles? Return me yes or no |
| 47 | Do eye corners have natural creases? Return me yes or no |
| 48 | Is there natural variation in lip texture? Return me yes or no |
| 49 | Are facial hair shadows realistic? Return me yes or no |
| 50 | Do glasses cast appropriate shadows? Return me yes or no |

Table 16: Questions list generated by Claude3.5-Sonnet (part2)

| No. | Question |
|---|---|
| 51 | Is there natural skin translucency? Return me yes or no |
| 52 | Are facial expressions emotionally consistent? Return me yes or no |
| 53 | Do neck muscles align naturally? Return me yes or no |
| 54 | Is there proper depth in smile lines? Return me yes or no |
| 55 | Are eye reflections consistent with scene lighting? Return me yes or no |
| 56 | Do facial features maintain proportion when moving? Return me yes or no |
| 57 | Is there natural skin aging present? Return me yes or no |
| 58 | Are hair strands individually visible? Return me yes or no |
| 59 | Do facial veins appear natural where visible? Return me yes or no |
| 60 | Is there consistent skin tone across transitions? Return me yes or no |
| 61 | Are nostril shapes symmetrical? Return me yes or no |
| 62 | Do ears have natural internal structure? Return me yes or no |
| 63 | Is there proper depth in nasolabial folds? Return me yes or no |
| 64 | Are eye bags and circles age-appropriate? Return me yes or no |
| 65 | Do facial piercings sit naturally? Return me yes or no |
| 66 | Is there natural variation in beard density? Return me yes or no |
| 67 | Are lip lines naturally defined? Return me yes or no |
| 68 | Do cheekbones have natural contours? Return me yes or no |
| 69 | Is there proper temple definition? Return me yes or no |
| 70 | Are eye whites naturally textured? Return me yes or no |
| 71 | Do facial scars appear authentic? Return me yes or no |
| 72 | Is there natural jaw definition? Return me yes or no |
| 73 | Are facial dimples naturally placed? Return me yes or no |
| 74 | Do eyebrow hairs have direction variation? Return me yes or no |
| 75 | Is there proper chin definition? Return me yes or no |
| 76 | Are facial freckles naturally distributed? Return me yes or no |
| 77 | Do eyelids have natural creases? Return me yes or no |
| 78 | Is there consistent skin shininess? Return me yes or no |
| 79 | Are facial tattoos properly embedded? Return me yes or no |
| 80 | Do smile lines appear natural? Return me yes or no |
| 81 | Is there proper forehead texture? Return me yes or no |
| 82 | Are eye corners naturally aged? Return me yes or no |
| 83 | Do facial muscles show proper definition? Return me yes or no |
| 84 | Is there natural lip symmetry? Return me yes or no |
| 85 | Are ear lobes naturally shaped? Return me yes or no |
| 86 | Do facial shadows have color variation? Return me yes or no |
| 87 | Is there proper nose bridge definition? Return me yes or no |
| 88 | Are facial pores consistently sized? Return me yes or no |
| 89 | Do eyebrows have natural thickness variation? Return me yes or no |
| 90 | Is there natural skin elasticity? Return me yes or no |
| 91 | Are facial creases movement-appropriate? Return me yes or no |
| 92 | Do teeth have natural translucency? Return me yes or no |
| 93 | Is there proper cheek coloring? Return me yes or no |
| 94 | Are eye bags naturally shadowed? Return me yes or no |
| 95 | Do facial features maintain proper scale? Return me yes or no |
| 96 | Is there natural skin undertone? Return me yes or no |
| 97 | Are facial expressions muscle-consistent? Return me yes or no |
| 98 | Do wrinkles have proper depth? Return me yes or no |
| 99 | Is there natural facial bone structure? Return me yes or no |
| 100 | Are skin textures consistently detailed? Return me yes or no |

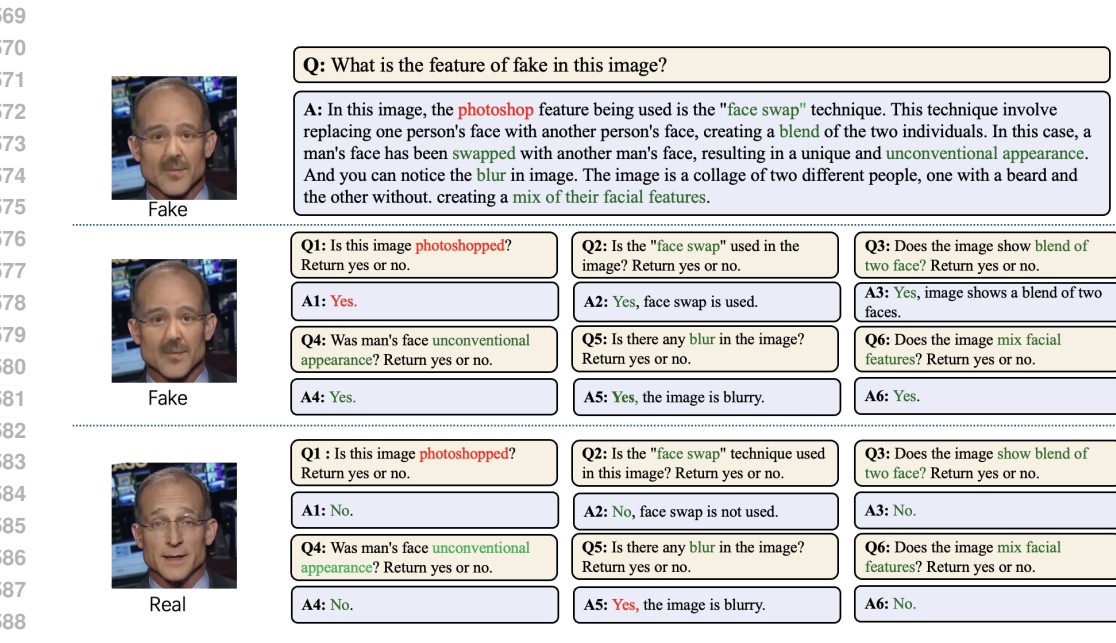

**Q:** What is the feature of fake in this image?

**A:** In this image, the photoshop feature being used is the "face swap" technique. This technique involve replacing one person's face with another person's face, creating a blend of the two individuals. In this case, a man's face has been swapped with another man's face, resulting in a unique and unconventional appearance. And you can notice the blur in image. The image is a collage of two different people, one with a beard and the other without. creating a mix of their facial features.

Fake

**Q1:** Is this image photoshopped? Return yes or no.

**A1:** Yes.

**Q4:** Was man's face unconventional appearance? Return yes or no.

**A4:** Yes.

**Q2:** Is the "face swap" used in the image? Return yes or no.

**A2:** Yes, face swap is used.

**Q5:** Is there any blur in the image? Return yes or no.

**A5:** Yes, the image is blurry.

**Q3:** Does the image show blend of two face? Return yes or no.

**A3:** Yes, image shows a blend of two faces.

**Q6:** Does the image mix facial features? Return yes or no.

**A6:** Yes.

Fake

**Q1 :** Is this image photoshopped? Return yes or no.

**A1:** No.

**Q4:** Was man's face unconventional appearance? Return yes or no.

**A4:** No.

**Q2:** Is the "face swap" technique used in this image? Return yes or no.

**A2:** No, face swap is not used.

**Q5:** Is there any blur in the image? Return yes or no.

**A5:** Yes, the image is blurry.

**Q3:** Does the image show blend of two face? Return yes or no.

**A3:** No.

**Q6:** Does the image mix facial features? Return yes or no.

**A6:** No.

Real

Figure 18: Bad sample of Pre-trained model (part3)

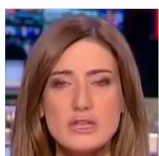

**USER:** Is this image real or fake? Tell the reason in detail?

**Pretrained MLLM:** This image appears real. It shows a woman speaking into a camera with her mouth wide open, possibly in an interview setting. Deepfake images often involve more discrepancies in facial features, lighting, and overall content when compared to a regular image with a genuine, real-world scenario. The image seems to capture a natural moment, so it does not suggest any deepfake content as a possibility.

**USER:** This image is fake, tell me the reasons.

**Pretrained MLLM:** The image is fake because it is a manipulated photo of a woman with a different background. The woman's face is placed onto a different body, creating an unnatural and unrealistic appearance. This manipulation is commonly done for entertainment, artistic purposes, or to create a humorous effect. The image does not depict a real-life scenario, and the altered elements make it a fake representation of a person.

**USER:** Is this image real or fake?

$\chi^2$**-DFD :** This image is fake. The image appears to be a deepfake as the woman's face appears to be distorted, with an unusual layout and unnatural skin tone. The facial features seem to be mismatched, and the overall appearance of the face is not consistent with a real person. And image show the obvious blending artifact.

Figure 19: Comparison between Pre-traind LLaVa and ours

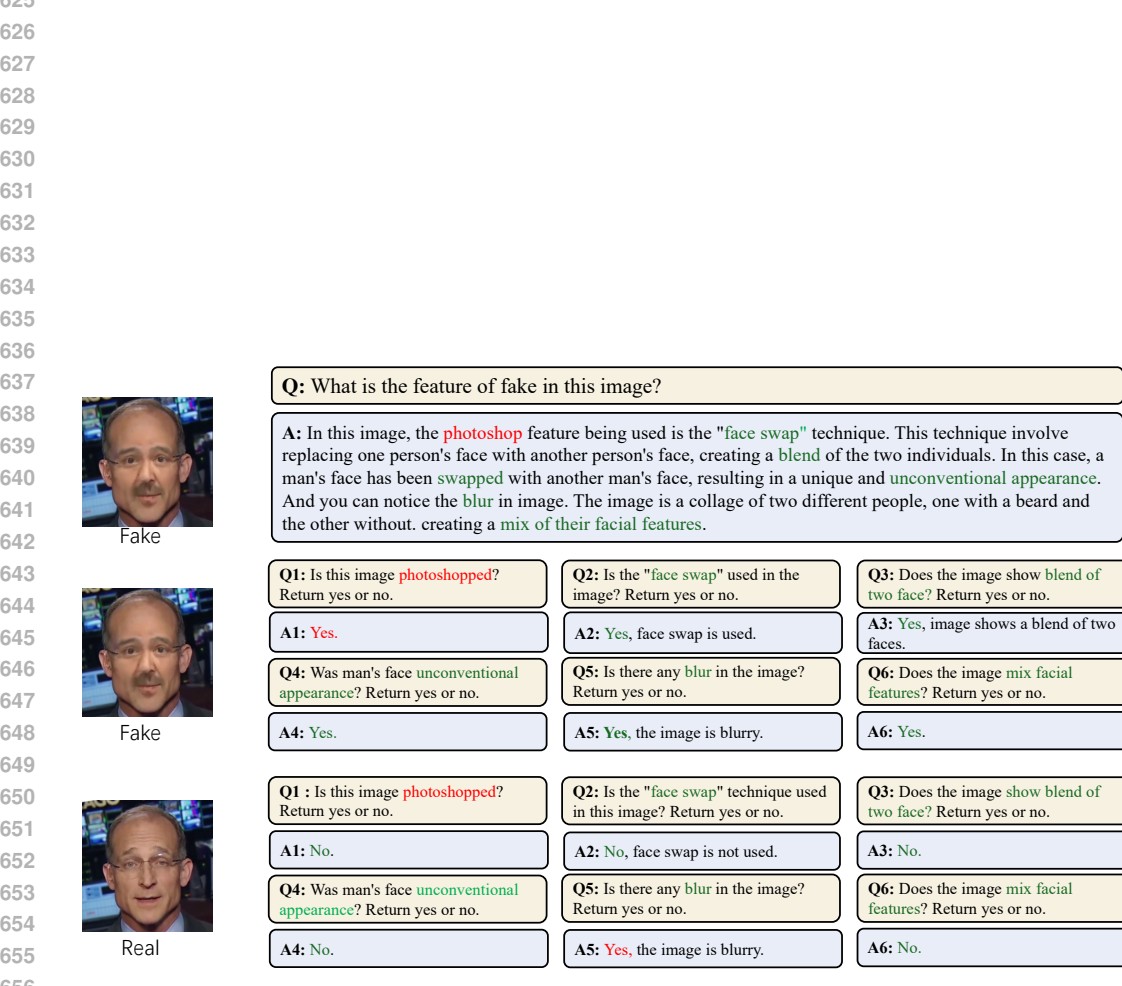

Figure 20: Feature related questions

