# OpenReview forum: "$\mathcal{X}^2$-DFD: A framework for e$\mathcal{X}$plainable and e$\mathcal{X}$tendable Deepfake Detection"
_ICLR.cc/2025/Conference — Submitted to ICLR 2025_

### Official Review · Reviewer_FRF6 · 2024-11-01

**Soundness:** 2
**Presentation:** 3
**Contribution:** 2
**Rating:** 3
**Confidence:** 5

**Summary:**

The authors propose a novel framework called X2-DFD, consisting of three core modules (i.e., Model Feature Assessment (MFA), Strong Feature Strengthening (SFS), and Weak Feature Supplementing (WFS)) for explainable deepfake detection. The results on several benchmark datasets demonstrate its effectiveness. However, the overall story and solution is interesting but with too much human workload and tricks while without enough technical contribution.

**Strengths:**

This paper adopts MLLMs for explainable deepfake detection. The cross-dataset performance seems good. The structure of the paper is clear.

**Weaknesses:**

1. The cross-manipulation performance presented in Table 2 is suboptimal despite using a trainable MLLM and a non-trainable GPT-4. Given the extensive parameters and pre-trained data leveraged, the proposed method cannot achieve SOTA performance. The capability of handling unseen attacks limits its applications in real-world scenarios.

2. X2-DFD’s explainablity is unsatisfactory for some samples, such as the left bottom sample in Figure 1, which outputs vague statements like “...with an unusual layout and unnatural skin tone...”. This description is oversimplified. It cannot provide sufficient information to users and offers less explanatory power than traditional methods like Grad-CAM (which can provide heatmaps). Is this a common issue across samples, and how is explainablity assessed (e.g., through user studies)?

3. The authors ignore prior work using LLMs for deepfake detection and over-claim their contribution (Lines 100-103)(r1,r2). The summary on Lines 067-075 misrepresents previous work [r1] by claiming“...fail to provide intuitive and convincing explanations behind predictions.”

[r1] FakeBench: Probing Explainable Fake Image Detection via Large Multimodal Models
[r2]Common Sense Reasoning for Deep Fake Detection

4. The authors do not present experiments on low-quality images or post-processed images.

5. The ablation study is insufficient. What would happen if WFS were applied to Strong Features or SFS to Weak Features? Or if both Strong and Weak Features used the same module?

6. The idea of using an external model (EDD in this paper) to support LLMs is not novel.

[r3] LISA: Large Language Instructed Segmentation Assistant
[r4] AnomalyGPT: Detecting Industrial Anomalies using Large Vision-Language Models

7. The reason for choosing LLaVA & GPT-4 is unclear. What would happen with different MLLMs, or different parameter configurations (e.g., LLaVA-13B/7B or GPT-3)? X2-DFD’s sensitivity to various MLLMs remains unexplored.

8. Writing issues:
(1) Citation inconsistencies, e.g., the reference in L136 is missing. ArXiv references, especially in L598-601 and L619-621, follow inconsistent formats.
(2) Missing percentage symbols in metrics reported on L175-180.
(3) Metric explanations occupy excessive space.
(4) Table 4 lacks clarity regarding which metrics are presented.


9 .During the reasoning process, the paper utilizes an external reasoning engine for analysis. However, if the results from this engine are subpar, it could significantly impact the model's performance. Unfortunately, the paper does not discuss this in detail（Especially, the training details of external expert models are not mentioned）. Additionally, in Table 3's ablation study, the minimal performance difference between the EDD only and final models suggests limited gains. Furthermore, using an untrained MLLM as a baseline for comparison seems unreasonable.

10. In the strong feature data selection phase, the paper considers only one metric, and similarly, it evaluates performance using just that metric. This lack of additional metrics related to accuracy (ACC) or natural language processing (NLP) interpretability may limit the comprehensiveness of the results.

11.The paper lacks a discussion on how common large models would perform in comparative experiments after fine-tuning, such as the effects of switching to different LLM base models, which is worth exploring further.

**Questions:**

see weakness for details.

---

> ### Author Response · Authors · 2024-11-23
> **Response to Reviewer FRF6 (Part 1/4)**
>
> **Q1. Given the extensive parameters and pre-trained data leveraged, the proposed method cannot achieve SOTA performance. The capability of handling unseen attacks limits its applications in real-world scenarios.**
>
> **R1:** Thank you for your comments; we would like to clarify that **our approach has already achieved SOTA performance in handling unseen attacks.**
>    - Results in **Table 1 (Cross-Dataset)** of the manuscript demonstrate that our method achieves SOTA performance across *all* datasets, with an **average improvement of 7.5%**.
>    - In **Table 2 (Cross-Manipulation)** of the manuscript, our method still achieves the best performance on three datasets and second-best on the remaining two, with an **average improvement of 6.9%**. This indicates robust generalization of our method across unseen manipulations.
>    - Furthermore, compare to the conventional detectors such as the second-best ProgressiveDet (2024 NIPS) in **Table1: Performance Comparison.** below, our method demonstrates **lower result variance and greater stability**. Stability under unseen scenarios is critical for real-world applications, and our results highlight this strength.
>    - Given these reasons, we believe that the statement *"The capability of handling unseen attacks limits its applications in real-world scenarios"* might not be accurate and fair. In contrast, the results reveal that our approach achieves the generally best generalization performance for both cross-dataset and cross-manipulation evaluations, as well as the lowest result variance and best stability.
>
> **Table1: Performance Comparison.**
>
> |Method|Venues|uniface|e4s|facedancer|fsgan|inswap|simswap|Avg|
> -|-|-|-|-|-|-|-|-
> CDFA|ECCV 2024|76.5|67.4|75.4|84.8|72.0|76.1|75.9
> ProgressiveDet|NeurIPS 2024|84.5|71.0|73.6|86.5|78.8|77.8|78.7
> Ours (*w/o CDFA*)|-|84.5|94.1|79.9|88.0|77.2|83.3|84.5
> Ours|-|85.2|91.2|83.8|89.9|78.4|84.9|85.6
>
> In summary, our method demonstrates strong and stable performance across various scenarios, including unseen attacks.
>
> REFERENCE:
>
> [1] Yan Z, Yao T, Chen S, et al. Df40: Toward next-generation deepfake detection. NeurIPS 2024.
>
> ---
> **Q2. X2-DFD’s explainability is oversimplified, which cannot provide sufficient information to users and offers less explanatory power than traditional methods like Grad-CAM (which can provide heatmaps). Is this a common issue across samples, and how is explainability assessed (e.g., through user studies)?**
> Here is the response with key points emphasized in **bold**:
> **R2:** Thank you for raising this important question about explainability. We would like to clarify the reasons for the improved explainability of our method from the following perspectives.
>
> 1. **Underlying Reasons for the Improvement of MLLM's Explainability by Our Approach:**
>    - The explanations provided by X2-DFD are derived from the **inherent strong feature of MLLMs**, ensuring the output explanations are **more reliable**, as we only leverage the features that (1) the model inherently understands and (2) have enough discrimination for explaining deepfakes. As a result, using unfamiliar features or less discriminative features for explanation) is largely minimized, thereby improving the reliability of the model's explainability.
>
> 2. **Detailed Comparison of Grad-CAM and Ours in Explainability:**
>    - Although many previous conventional detectors use Grad-CAM for demonstrating the model's explainability, the visualization can only provide a **rough hint**, *e.g.,* highlighting the whole face region, but fails to further point out and explain the more fine-grained and detailed artifacts within the located region.
>    - In contrast to Grad-CAM, our approach is able to provide **more fine-grained explanation** about the specific forgery artifacts such as unnatural facial layout, blurry eyes, etc. Furthermore, we also leverage conventional detectors (EDDs) to further supplement the explainability of weak features.
>
> 3. **Explainability Assessment through Human/User Studies:**
>    - To assess and compare the explainability of our approach with other MLLMs, **we have conducted a human study** involving *well-educated participants* who were provided with *clear guidelines*. Participants were asked to evaluate and compare explanations generated by different MLLMs to determine which they found most informative. We believe this assessment can provide a reasonable illustration to show that our approach has better explainability to humans.
>    - Additionally, our human-study experiment has undergone **institutional review board (IRB) approval**, adhering to ethical standards. Relevant documentation, including ethical review details and approvals, will be provided post-blind review, along with additional experimental details. We believe this transparency may further address and alleviate concerns regarding the Flag for Ethics Review.

---

> ### Author Response · Authors · 2024-11-23
> **Response to Reviewer FRF6 (Part 2/4)**
>
> **Q3. The authors ignore prior work using LLMs for deepfake detection and over-claim their contribution (Lines 100-103)(r1,r2). The summary on Lines 067-075 misrepresents previous work [r1] by claiming“...fail to provide intuitive and convincing explanations behind predictions.”
> [r1] FakeBench: Probing Explainable Fake Image Detection via Large Multimodal Models
> [r2]Common Sense Reasoning for Deep Fake Detection**
>
>
> **R3:** We appreciate your valuable feedback and the opportunity to clarify and improve our work. Below, we address the concerns raised:
>
> 1. **Clarification of Scope and Context:**
>    - To avoid any misunderstanding, we would like to clarify that lines **47-75 in the paper** evaluate deepfake detection in the context of **traditional models**. Discussions of **LLM-related work**, including the ones you mentioned ([R1], [R2]), are presented later in the text (lines **125-140**).
>    - As stated in our paper: *“To our knowledge, we are the first to **systematically assess the inherent capabilities** of MLLMs specifically in deepfake detection.”*
>
> 2. **Relation to Prior Work and Scope Exclusion:**
>    - We acknowledge existing efforts using LLMs for deepfake detection, such as:
>      1. **FakeBench** [1]: Evaluates **full (natural) image synthesis** using MLLMs.
>      2. **Common sense reasoning** [2]: Utilizes human-labeled datasets to explore how textual explanations can improve detection performance, generalization, and interpretability.
>      3. **Can chatgpt detect deepfakes? [3]**: Investigates the capabilities of multimodal large language models for deepfake detection.
>    - Our work provides a distinct contribution by **accessing the inherent capabilities of MLLMs and deeply analyzing** in the specific task of **face forgery detection**.
>     - As noted in the footnote of our paper, our scope is explicitly limited to **face forgery detection** and does not extend to **full-image synthesis** tasks, such as those addressed in [1]. This distinction in focus explains why [1] was not a primary consideration in our discussion. However, we acknowledge its relevance and have revised the manuscript (lines 131–132) to make this distinction more rigorous.
>
> 3. **Positioning of Our Contribution:**
>    - While we are not the first to use MLLMs for deepfake detection, we are the first to systematically **assess the inherent capabilities of MLLMs in face forgery detection**.
>    - This analysis provides a significant contribution to understanding the potential of MLLMs and demonstrates the necessity of leveraging **strong features from MLLMs** within our proposed framework. We believe this step is crucial for advancing the field.
>
> We have revised the manuscript for clarity and updated Lines 131-133 to cite FakeBench.
>
> REFERENCE:
>
> [1] Li Y, Liu X, Wang X, et al. FakeBench: Uncover the Achilles' Heels of Fake Images with Large Multimodal Models. ArXiv 2024.
>
> [2] Zhang Y, Colman B, Guo X, et al. Common Sense Reasoning for Deepfake Detection. ECCV 2024.
>
> [3] Jia S, Lyu R, Zhao K, et al. Can chatgpt detect deepfakes? a study of using multimodal large language models for media forensics. CVPRW 2024.
>
> ---
> **Q4. The authors do not present experiments on low-quality images or post-processed images.**
>
> **R4**.Thank you for this valuable suggestion. We would like to clarify that experiments on low-quality and post-processed images, including **Gaussian blur, block-wise distortion, contrast changes, and JPEG compression**, are presented in **Appendix A.3: Evaluation of Robustness Against Unseen Perturbations**. These results demonstrate our model's strong robustness in such scenarios.
>
> To improve clarity, we have added the content in Lines 533-536 of the updated manuscript, helping readers locate the detailed results in the appendix.

---

> ### Author Response · Authors · 2024-11-23
> **Response to Reviewer FRF6 (Part 3/4)**
>
> **Q5. The ablation study is insufficient. What would happen if WFS were applied to Strong Features or SFS to Weak Features? Or if both Strong and Weak Features used the same module?**
>
> **R5:** Thank you for raising this point. we would like to clarify the roles of Strong Feature Strengthening (SFS) and Weak Feature Supplementation (WFS).
>
> 1. **Roles of SFS and WFS:**
>    - **SFS** enhances the model's **strong features** by focusing on areas where it already performs well.
>    - **WFS** supplements **weak features** using additional information to improve areas where the model struggles.
>
> 2. **Impact of Applying SFS to Strong vs. Weak Features:**
>    - **SFS on Strong Features:** This helps the model better leverage its strengths. As shown in **Figure 6**, this method boosted recognition performance for the **Top 1/3 features** by **28.6%**.
>    - **SFS on Weak Features:** Applying SFS to weak features is ineffective because these features lack reliable annotations. Fine-tuning with unreliable data would yield poor performance and lead to untrustworthy outputs, similar to the **No SFS baseline**（random）.
>
> 3. **Exploration of Suggested Configurations:**
>    - We experiment with applying SFS alongside EDD for text annotation. As shown in **Table 2: Performance Comparison**, this approach resulted in only a minor performance gain (+1.19%) and proved less effective compared to our framework.
>
> **Table2: Performance Comparison.**
>
> | **Module**             | **CDF-v2** | **DFDCP** | **DFDC** | **DFD** | **Uniface** | **E4S** | **FaceDancer** | **FSGAN** | **Inswap** | **SimSwap** | **Avg** |
> | ---------------------- | ---------- | --------- | -------- | ------- | ----------- | ------- | -------------- | --------- | ---------- | ----------- | ------- |
> | **SFS**                | 83.3       | 82.0      | 79.2     | 91.4    | 84.5        | 94.1    | 79.9           | 88.0      | 77.2       | 83.3        | 84.29   |
> | **EDD Annotate + SFS** | 83.0       | 81.4      | 79.6     | 89.4    | 86.4        | 95.9    | 81.8           | 89.8      | 81.2       | 86.3        | 85.48   |
>
> ---
>
>
>
> **Q6. The idea of using an external model (EDD in this paper) to support LLMs is not novel.
> [r3] LISA: Large Language Instructed Segmentation Assistant
> [r4] AnomalyGPT: Detecting Industrial Anomalies using Large Vision-Language Models**
>
> **R6:** Thank you for pointing out this observation. Below, we clarify the role of EDD in our work and its broader significance:
>
>
> 1. **Unique Contribution of Our Approach:**
>    - Thanks for mentioning [r3] and [r4], which are two studies showing the promise of MLLMs in using extern models for enhanced performance, in segmentation and industrial anomaly detection tasks. However, the novelty and ignorable contribution of our work is reflected below:
>      - Utilizing extern models is a task-specific design tailored to the *deepfake detection field*. Given numerous well-performance expert conventional smaller detectors available currently, it is still questionable whether and how these smaller detectors can benefit the MLLMs for a comprehensive and more robust detection, creating "1+1>2" results. To our knowledge, there is a lack of exploration of the specific and effective strategies to address this, making the question still challenging and unexplored. To address this issue, we have proposed a reasonable and effective framework for implementation. For these reasons, our strategy is novel and has a positive contribution to the current field.
>      - Utilizing extern models accurately meets the needs in our case, where we found the MLLMs lack several capabilities in detection based on the **rigorous analysis of the limitations of MLLMs.** Therefore, using extern models in our framework is suitable and can supplement the weak abilities of the original MLLMs.
>
>
> 2. **Broader Perspective on Tool Integration:**
>    - On a macro level, the use of external models reflects the **expandability of our framework**, which is inspired by the concept of **tool learning** in LLM. This paradigm has been explored across various fields and represents a **key strength of LLMs**. To some extent, we may also have been **inspired by these advancements** and applied this concept in a way that effectively enhances deepfake detection in our framework.
>
> By integrating EDD, we not only address practical challenges in our task but also demonstrate how such tool-based extensibility can be harnessed in a focused and impactful manner. Thank you for your comments, which have allowed us to articulate this point more clearly!

---

> ### Author Response · Authors · 2024-11-23
> **Response to Reviewer FRF6 (Part 4/4)**
>
> **Q7. The reason for choosing LLaVA & GPT-4 is unclear. What would happen with different MLLMs, or different parameter configurations (e.g., LLaVA-13B/7B or GPT-3)? X2-DFD’s sensitivity to various MLLMs remains unexplored. The paper lacks a discussion on how common large models would perform in comparative experiments after fine-tuning, such as the effects of switching to different LLM base models**
>
> **R7:** Thank you for raising this important question. Below, we clarify our rationale and address your concerns:
>
> 1. **Rationale for Selecting GPT-4 and LLaVA:**
>    - We select **GPT-4** and **LLaVA** because they are **representative models** in their respective categories:
>      - **GPT-4** serves as a state-of-the-art **closed-source model**, offering advanced capabilities.
>      - **LLaVA** is a leading **open-source model**, widely adopted in research for its accessibility and versatility.
>    - Our aim was to evaluate how our framework can **adapt to and leverage different MLLMs**, rather than focusing solely on comparing specific models.
>
> 2. **Additional Experiments:**
>    To further address your concerns, we conduct additional experiments with different MLLMs, parameter configurations, and LLMs for generating questions:
>    - **Different LLMs for Question Generation:** Using various LLMs in MFA consistently demonstrates that our framework performs robustly, irrespective of the LLM used for question generation.
>    - **Different Parameter Configurations:** As the model size increases (e.g., LLaVA-7B to LLaVA-13B), the framework’s performance improves proportionally, benefiting from the enhanced capabilities of larger MLLMs.
>    - **Different MLLMs for Fine-tuning:** The results indicate that our framework does not rely on a specific MLLM, performing strongly across various MLLMs tested.
>
> We have included additional details and results in **Appendix 5** of the updated manuscript.
>
> Thank you again for your insightful suggestion. We hope this discussion addresses your concerns and provides further clarity.
>
> **Q8. Writing issues: (1) Citation inconsistencies, e.g., the reference in L136 is missing. ArXiv references, especially in L598-601 and L619-621, follow inconsistent formats. (2) Missing percentage symbols in metrics reported on L175-180. (3) Metric explanations occupy excessive space. (4) Table 4 lacks clarity regarding which metrics are presented.**
>
> **A8** We have addressed the writing issues as follows:
>
> 1. Added the missing reference in L136, ensured consistent formatting for ArXiv references (L598–601, L619–621), included missing percentage symbols (L175–180), and clarified in Appendix 2 that Table 4 uses AUC as the evaluation metric.
>
> 2. Condensed the metric explanations to improve brevity while retaining clarity.
>
> These revisions enhance the manuscript’s readability and precision.
>
> **Q9. During the reasoning process, the paper utilizes an external reasoning engine for analysis. However, if the results from this engine are subpar, it could significantly impact the model's performance. Unfortunately, the paper does not discuss this in detail（Especially, the training details of external expert models are not mentioned）. Additionally, in Table 3's ablation study, the minimal performance difference between the EDD-only and final models suggests limited gains. Furthermore, using an untrained MLLM as a baseline for comparison seems unreasonable.**
>
> **R9:** Thank you for your comments. We would like to address your concerns as follows:
>
> 1. **Impact of EDD and Analysis in A.2:**
>    - We acknowledge the potential impact of EDD on the overall performance. To address this, we conduct a detailed study in **Appendix A.2: Feature Supplementing Analysis**, where we experiment with multiple EDD models and derive **three criteria**（line 813-819）for selecting suitable EDDs.
>     - While EDD (e.g., CDFA) demonstrates strong standalone performance, the **average performance improvement from 75.9% to 85.6% (a 9.7% absolute gain)** in **Table 2** highlights the significant value of integrating EDD with our model. This improvement underscores the complementary role of our framework in leveraging EDD outputs for better overall results generalization and robustness.
>
> 2. **Training Details of External Expert Models:**
>     - The training details of the EDD models are provided in **Appendix A.2** (lines 821–825, 830–833, and 862–863).
>     - Additionally, the training process for the WFS module (which is responsible for learning how to integrate EDD into the framework) is detailed in **Section 4.2.4** (lines 362–392).
>
> 3. **Regarding Pre-trained MLLM as a Baseline:**
>    - The pre-trained MLLM serves as the **foundation** of our framework, and all improvements are built upon it. While this is the most basic module, we also provide additional baselines such as **No SFS**, which can be viewed as another comparison point for performance without specific enhancements.
>
> Thank you again for your suggestions!

---

> ### Author Response · Authors · 2024-11-23
> **Response to Reviewer FRF6 (Part 4/4 continued)**
>
> **Q10. In the strong feature data selection phase, the paper considers only one metric, and similarly, it evaluates performance using just that metric. This lack of additional metrics related to accuracy (ACC) or NLP interpretability may limit the comprehensiveness of the results.**
>
> **A10**: Thank you for these comments. We would like to address your concerns as follows:
>
> 1. **Rationale for the Strong Feature Data Selection Metric:**
>    - The metric used in the strong feature data selection phase is specifically designed to identify **features that are highly discriminative for distinguishing real and fake images** in the training dataset.
>     - handle data imbalance in the training dataset (fake:real=4:1), we compute **balance accuracy (average of real image accuracy and fake image accuracy)** to ensure fair evaluation of features.
>
> 2. **Relationship to ACC:**
>
>    - This metric ensures that the selected strong features are not biased toward one class, and when the number of real and fake images is equal, the **balance accuracy** equals the overall ACC.
>
> 3. **Relevance to the Task vs. NLP Metrics:**
>    - Our focus is on selecting **discriminative features** that effectively separate real and fake images. Metrics such as perplexity or other NLP-related interpretability measures are not directly relevant to our deepfake detection task (a vision task).
>
> We hope this explanation clarifies the reasoning behind our metric choice, its relationship to ACC, and its alignment with the goals of our task. Thank you for your thoughtful suggestion, which helps us better articulate these points in our work.

---

> > ### Author Response · Authors · 2024-11-25
> > **Appreciation for Review and Request for Feedback**
> >
> > Dear Reviewer FRF6,
> >
> > We want to convey our sincere appreciation for the valuable insights and suggestions you provided regarding our work.
> >
> > We have made efforts to address the concerns and queries you raised during the rebuttal process. It would be immensely helpful to receive feedback on whether our response effectively alleviated any doubts you may have had. Your feedback is crucial to enhancing the quality of our work.
> >
> > Recognizing the demands of your busy schedule, we genuinely appreciate your contribution to the refinement of our manuscript. As the end of the rebuttal period is approaching, we eagerly await your reply before the end.
> >
> > Once again, thank you for your time and consideration.
> >
> > Best regards,
> >
> > Authors

---

> > > ### Comment · Reviewer_FRF6 · 2024-12-01
> > > **Keep My Rating**
> > >
> > > Thanks for your detailed response, which addresses partial concerns. In consideration of the lack of technical depth and contribution, I still keep my original rating. Thanks.

---

### Official Review · Reviewer_Umwy · 2024-11-02

**Soundness:** 3
**Presentation:** 3
**Contribution:** 3
**Rating:** 8
**Confidence:** 4

**Summary:**

The paper proposes X2-DFD, a novel framework that utilizes Multimodal Large Language Models (MLLMs) for explainable and extendable DeepFake Detection. The basic idea of the paper is quite interesting. And the paper seems to the first to systematically assess the inherent capabilities of MLLMs specifically in deepfake detection, which is valuable.

**Strengths:**

1.The paper first systematically assesses the inherent capabilities of MLLMs specifically in deepfake detection, and has found that MLLMs have varying discriminating capabilities on different forgery features.
2.It proposes a novel approach to fine-tune the MLLM to make it better adaptive with the deepfake detection task.
3.Besides MLLM, it integrates external dedicated detectors (EDDs) to fill the gap where MLLMs show limitations.

**Weaknesses:**

Although interesting to see MLLM being applied in improving Deepfake detection performance, I still wonder if the interpretability brought by  MLLM is reliable. Therefore I have two major concerns:
1.	The explainability of large language models (LLMs) like GPT, BERT, and similar models is a complex and evolving area. Generally, LLMs are not inherently transparent or easily interpretable due to their massive size and the intricacy of their neural architectures. So, how do you ensure the explainability brought by LLMs is “real” explainability instead of merely providing some texts describing why an image is fake or not.
2.	Based on my first point, how would you evaluate the explainability of the proposed method? There seems to be a lack of metric or ground-truth involved in your research. I wonder if human evaluation is fairly enough to evaluate the explainability since human subjects involved in the experiments also have no real knowledge about how the fake data was created and why it is fake. By the way, the details of the human experiment are not provided.
In my opinion, it will be more promising to integrate the knowledge about the deepfake creation and the true difference between real and fake data into the detection stage in order to improve the detector’s explainability. Nevertheless, I still think it is a good paper which provides a good trial on solving deepfake detection with MLLMs.

**Questions:**

The extendability of the proposed method should be further explained. Most efforts of the paper are on the explainability. I do not clearly see what you mean by “extendable”.

---

> ### Author Response · Authors · 2024-11-23
> **Response to Reviewer Umwy**
>
> **Q1 How do you ensure the explainability brought by LLMs is "real" explainability instead of merely providing some texts describing why an image is fake or not?**
>
>
> **A1:** Thank you for this thoughtful question. We completely agree that **LLMs are not inherently transparent or easily interpretable**. This is why we designed the **Model Feature Assessment (MFA)** module to systematically evaluate the discriminative capabilities of MLLMs, enhancing the strong discriminative features and supplementing the weak. Below is the detailed explanation of how we ensure **real explainability**:
>
> 1. **Assessing Real Discriminative Features:**
>    - The proposed MFA module evaluates the vast capabilities of MLLMs to identify **features that are truly known** and discriminative for distinguishing real and fake content. This ensures that the explanations are grounded in meaningful and relevant features rather than arbitrary text.
>
> 2. **Enhancing Feature Understanding:**
>    - For strong forgery-related features, we apply **Strong Feature Strengthening (SFS)** to improve the model’s ability to recognize and explain these features. As shown in Figure 6, **Top 1/3 features see a relative improvement of 28.6%**, demonstrating enhanced capability.
>
> 3. **Ensuring Accuracy for Weak Features:**
>    - For weak features, we incorporate an external detector (**EDD**) to ensure the accuracy of feature extraction and interpretation. This extension allows the framework to provide reliable and evidence-based explanations for weak features.
>
> By combining these approaches, our framework ensures that the explanations are grounded in real, identifiable features rather than merely generating arbitrary text descriptions. This comprehensive design ensures that **explanations are evidence-based and meaningful**. Thank you again for your question, which allowed us to clarify this critical aspect of our work!
>
>
> **Q2. Based on my first point, how would you evaluate the explainability of the proposed method? There seems to be a lack of metric or ground truth involved in your research. I wonder if human evaluation is fairly enough to evaluate the explainability since human subjects involved in the experiments also have no real knowledge about how the fake data was created and why it is fake.**
>
> **A2:** Thank you for raising this valuable question. Here are our responses:
>
> 1. **Challenges of Ground Truth for Explainability:**
>    The lack of **detailed annotations of specific forgery artifacts** makes it challenging to accurately quantify and evaluate explainability. To our knowledge, there are no existing studies proposing quantitative evaluation metrics for explainability that can be directly applied to our framework.
>
> 2. **Human Study Design and Reliability:**
>    - While the absence of detailed annotations limits precise quantification, human evaluators offer a practical solution for **quantitative assessment**, as detailed in **Section 5.5** and **Appendix A.4**.
>    - Although human assessments may not be completely objective, our human study is designed to ensure **reliability**:
>      - We included **well-educated participants** who were provided with **clear guidelines** to ensure consistent and reliable results.
>      - Additional details of the human study, including the methodology and guidelines, have been added to **Appendix A.4** and highlighted in **red** (lines 928-933). Due to the double-blind review process, specific details such as **ethical approval documents** will be included in the final version.
>      - We believe this carefully designed human study offers a reliable way to evaluate explainability in the absence of detailed forgery artifact annotations.
>
> 3. **Integrating Deepfake Prior Knowledge to Improve Explainability:**
>    Thank you for this constructive suggestion. We agree that incorporating **prior knowledge of deepfake** into the framework is a promising direction worth exploring. While we do not currently have a detailed plan, we believe this idea could potentially be integrated into the process between **MFA and SFS** stages, enhancing the model’s ability to explain its reasoning. We consider this an interesting avenue for future work and plan to explore it further in subsequent studies.
>
> We greatly appreciate your insightful comments, which have helped us refine and improve our framework.
>
>
>
> **Q3. The extendability of the proposed method should be further explained. Most efforts of the paper are on explainability. I do not clearly see what you mean by "extendable".**
>
> **A3:** Thank you for your question. By **"extendable,"** we mean that our framework can seamlessly integrate external dedicated detectors (EDDs) by constructing VQA datasets and training the model to utilize EDD outputs to supplement weak features. A system analysis of this extendability is provided in **Appendix 2**. We have added a **Content Structure of the Appendix** in **Section 5** for clearer guidance.

---

> ### Author Response · Authors · 2024-11-25
> **Appreciation for Review and Request for Feedback**
>
> Dear Reviewer Umwy,
>
> We want to convey our sincere appreciation for the valuable insights and suggestions you provided regarding our work.
>
> We have made efforts to address the concerns and queries you raised during the rebuttal process. It would be immensely helpful to receive feedback on whether our response effectively alleviated any doubts you may have had. Your feedback is crucial to enhancing the quality of our work.
>
> Recognizing the demands of your busy schedule, we genuinely appreciate your contribution to the refinement of our manuscript. As the end of the rebuttal period is approaching, we eagerly await your reply before the end.
>
> Once again, thank you for your time and consideration.
>
> Best regards,
>
> Authors

---

### Official Review · Reviewer_37wm · 2024-11-03

**Soundness:** 3
**Presentation:** 4
**Contribution:** 3
**Rating:** 6
**Confidence:** 5

**Summary:**

In this paper, the authors use a multi-modal large model to enhance the interpretability of face forgery detection. To construct an effective fine-tuning dataset, they propose first ranking various forgery features and then building a corresponding VQA dataset based on the top-ranked features. Additionally, the detection results from smaller models are fed into the LLM as extra guidance.

**Strengths:**

1. The recognition effect has been greatly improved, especially for cross-dataset scenarios.
2. It is the first exploration to design a novel MLLMs-based framework for explainable forgery detection.
3. The experiments are thorough and the comparisons are comprehensive, effectively demonstrating the validity of the proposed method.
4. The paper is well-organized and easy to understand.

**Weaknesses:**

1. It is hard to believe the simple fine-tuning of multi-modal large models typically can achieve such satisfactory results. The public source code can help prove its reproducibility.
2. Directly using the prediction probabilities of the smaller model as text input for the LLM seems unreasonable, as LLMs lack numerical reasoning capabilities.
3. Since Pretrained LLMs lack fundamental knowledge about true/false classification, ranking based on relevance generated by Pretrained LLMs may not be meaningful. It may be that random selection of other questions has a similar effect. Can the author provide further analysis?
4. Authors only select the specific LLM and MLLM models in the framework. How to ensure the generality and reasonability of the conclusion?

**Questions:**

1. Only the LLaVa model is chosen as the typical MLLM. Do most MLLMs contain similar properties? The author should point this out.
2. How can the commonality of the definition of forgery features be ensured? Authors utilized the GPT-4o to generate a questions list, how about other LLM models? Different LLM models may generate different question lists to extract forgery-related features.
3. In TABLE 2. Why the proposed method is inferior to comparison methods with fsgan and inswap generation models? Please give a deeper theoretical analysis.
4. In TABLE 3, the GCS concept was not explained clearly, and the ablation study settings are somewhat confusing.
5. To ensure the integrity of the experiment, authors also could provide in-domain results in the FF++ dataset.

---

> ### Author Response · Authors · 2024-11-23
> **Response to Reviewer 37wm (Part 1/2)**
>
> **Q1. It is hard to believe the simple fine-tuning of multi-modal large models typically can achieve such satisfactory results. The public source code can help prove its reproducibility.**
>
> **A1:** Thank you for your positive feedback on our work! The effectiveness of our model can be attributed to two key strategies:
>
> - **Focusing on Strong Discriminative Forgery Features**: our approach enables the MLLMs to focus more on **strong discriminative forgery-related features** while eliminating distractions from weaker discriminative features. Additionally, after fine-tuning the MLLM with the strong feature, the model's ability to recognize strong features is further improved (as shown in **Figure 6**), enhancing the overall detection performance.
>
> - **Integrating with EDD to Supplement the Weak Features**: in addition to fine-tuning the MLLM using its strong forgery-related features, we also consider leveraging the EDD to supplement its "weakness". Specifically, by fine-tuning MLLMs with EDD's outputs, we aim to encourage the MLLMs to effectively leverage **accurate information** from EDD, addressing its weaker capability and further strengthening its overall capabilities.
>
> These strategies work together to ensure the model achieves satisfactory performance. As mentioned in the main text, we are committed to releasing our code for reproducibility **once the paper is accepted**. Our fine-tuning implementation is based on the official **LLaVA** fine-tuning implement, ensuring both **reproducibility and reliability**.
>
> **Q2. Directly using the prediction probabilities of the smaller model as text input for the LLM seems unreasonable, as LLMs lack numerical reasoning capabilities.**
> **A2:** Thank you for your insightful feedback, which allows us to clarify this important aspect of our framework.
>
> - **Not Relying on Numerical Reasoning Capabilities**:
>    We agree that large language models generally lack strong numerical reasoning capabilities. However, our approach does *NOT* explicitly rely on the numerical reasoning ability. Instead, our framework is designed to **learn how to effectively *use* the accurate information from EDD as supplementary**. Below, we have conducted experiments to confirm that the MLLM lacks numerical reasoning capabilities but has enough learning capabilities from the numerical data.
> - **Experimental Verification**:
>    To validate it, we have conducted experiments in *Table 1* (detailed in the revised version **Table 9**) comparing performance with and without training. In the **"no train + infer"** row, where the *Blend Score* is directly added to the prompt without *prior training*, the model struggles to effectively understand and utilize the scores, as it relies solely on its numerical reasoning capabilities.
>
> **Table 1 Compare with Numerical Reasoning Capabilities in LLM and our framework.**
>
> |Configuration|CDF-v2|DFD|DFDC|DFDCP|Avg|
> -|-|-|-|-|-
> no train + infer|0.8171|0.9062|0.7906|0.8134|0.8318
> **train + infer**|**0.9062**|**0.9232**|**0.8300**|**0.8873**|**0.8867**
>
> We appreciate your thoughtful comments and hope this explanation clarifies our approach.
>
> **Q3. Since Pretrained LLMs lack fundamental knowledge about true/false classification, ranking based on relevance generated by Pretrained LLMs may not be meaningful. It may be that random selection of other questions has a similar effect. Can the author provide further analysis?**
>
> **A3:** Thank you for your thoughtful feedback. We would like to clarify that the MLLMs have **partial** fundamental knowledge about the deepfake detection tasks, as shown in **Table 13**, where certain features exhibit strong discriminative power. A detailed analysis of their strengths and weaknesses is provided in Section 3.3.
>
> To further validate why the generated questions are meaningful, we provide the following clarifications:
>
> - **Utilizing the Strong Discriminative Capabilities of MLLMs for Fine-tuning:** Pretrained MLLMs show potential in strong discriminative features. Fine-tuning amplifies this strength by focusing on these strong discriminative forgery features, helping the model prioritize them while reducing its attention to weaker, less relevant ones, akin to feature selection.
>
> - **Random Selection vs. SFS:**
>      - Randomly selecting questions, as suggested, the line **No SFS** simulating random selection in **Table 3**, where not select Strong Feature to construct datasets for finetune. Leading to suboptimal performance.
>      - Furthermore, by focusing on Strong Features, the model is guided to provide reasons based on these highly discriminative features. In contrast, using randomly selected features often results in less credible explanations.
>
> **Table 2 Random Selection vs. SFS.**
>
> |Method|CDF-v2|DFD|DFDC|Uniface|Avg|
> -|-|-|-|-|-
> Random(no SFS)|79.0|88.9|77.8|82.3|82.0
> Selected (SFS)|83.2|91.4|82.0|84.5|85.3
>
> Thank you once again for your valuable comments and suggestions, which have greatly helped us refine and improve our work.

---

> ### Author Response · Authors · 2024-11-23
> **Response to Reviewer 37wm (Part 2/2)**
>
> **Q4. Authors only select the specific LLM and MLLM models in the framework. How to ensure the generality and reasonability of the conclusion? Only the LLaVa model is chosen as the typical MLLM. Do most MLLMs contain similar properties? The author should point this out.**
>
> **A4:** Thank you for your valuable question. We initially select LLaVA and GPT-4 due to their popularity and widespread adoption in various applications.
>
> To address your concern and ensure the generality of our framework, we expand our experiments as follows:
>
> - **For question generation (MFA):** We employ a different LLM, **Claude 3.5-Sonnet**, to generate the question list.
> - **For fine-tuning (SFS and WFS):**
>   1. We test different model sizes of the same MLLM, such as **LLaVA-7B** and **LLaVA-13B**.
>   2. We evaluate other MLLMs, including **Phi-3-Vision**.
>
> These additional experiments, detail in **Appendix 5**, confirm that our framework is not dependent on specific LLMs or MLLMs. Furthermore, the results indicate that advancements in MLLMs are likely to enhance performance even further.
>
> **Table 3 Experiments on different LLMs/MLLMs.**
>
> |Variant|CDF-v2|DFDCP|DFDC|DFD|Uniface|e4s|Facedancer|FSGAN|Inswap|Simswap|Avg|
> -|-|-|-|-|-|-|-|-|-|-|-
> GPT4o + Phi-3-vision|88.6|87.1|83.5|90.9|81.8|77.5|78.8|85.7|77.5|80.6|83.2
> GPT4o + LLaVa-7B|90.3|89.7|83.5|92.5|85.2|91.2|83.8|89.9|78.5|84.9|87.0
> Claude3.5-Sonnet + LLaVa-7B|88.8|88.5|82.6|92.7|84.6|90.1|83.8|89.7|79.5|85.6|86.6
> GPT4o + LLaVa-13B|91.3|90.3|83.4|92.5|86.0|92.5|84.5|91.0|80.6|85.4|87.8
>
> We appreciate your insightful feedback, which has allowed us to strengthen the comprehensiveness and generality of our work.
>
> **Q5. How can the commonality of the definition of forgery features be ensured? Authors utilized the GPT-4o to generate a questions list, how about other LLM models?**
>
> **A5:** Thank you for your thoughtful question. We address the commonality of forgery features and the performance of other LLMs as follows:
>
> - **Commonality of Forgery Features**
>   1. **Question Reliability via Prompt Design:**
>      We guide LLMs using a carefully designed prompt (Figure 4: FMA Question Generation) to ensure that generated questions are:
>      - **Clear:** Focused on forgery-related features.
>      - **Diverse:** Avoiding redundancy and ambiguity.
>      - **Broad Coverage:** Adjusted to ensure a wide range of forgery features.
>   2. **Support for Human Verification:**
>      Human verification is supported to ensure reliability in our framework. Most questions generated by the LLMs are accurate and meet requirements without errors, as demonstrated in **Table 13** and **Table 14**.
>
> - **Performance of Other LLM Models:**
>   Other LLMs, such as Claude 3.5-Sonnet, also generate diverse and reliable questions (**Table 15**, **Table 16**) and contribute to performance improvements within our framework (**Table 6**).
>
> Thank you again for your valuable feedback, which has helped us enhance our work.
>
> **Q6. In TABLE 2. Why the proposed method is inferior to comparison methods with fsgan and inswap generation models? Please give a deeper theoretical analysis.**
>
> **A6:** Thank you for your insightful question. While the performance of the proposed method is slightly inferior to the other baselines on FS-GAN and InSwap-generated fake data, the results are still quite close. Moreover, we would like to point out that traditional methods exhibit a larger variance in their performance, whereas our method shows robust performance and lower variance on different forgeries and largely outperforms other traditional approaches by 7% points in terms of average AUC performance.
>
> It is possible that traditional methods focus more on specific types of forgeries (more like "specialist"), leading them to perform well on certain forgery patterns but not generalize as effectively, while our method is more robust and can generalize across a broader range of forgeries (like "generalist"). Our framework provides an effective strategy for combining MLLMs and conventional approaches, thereby demonstrating superior results.
>
> **Q7. In TABLE 3, the GCS concept was not explained clearly, and the ablation study settings are somewhat confusing.**
>
> **A7:** Thank you for pointing this out.
>
> - **WCS** refers to **Weak Capability Strengthening**, aligned with **Weak Feature Strengthening (WFS)**.
> - **GCS** refers to **Good Capability Strengthening**, aligned with **Strong Feature Strengthening (SFS)**.
>
> We have updated these terms for clarity and revised **Lines 478-479** and **492-493** in **Section 5.4** of the manuscript.
>
> **Q8. To ensure the integrity of the experiment, authors also could provide in-domain results in the FF++ dataset.**
>
> **A8:** Thank you for your suggestion. We have supplemented the in-domain experimental results on the **FF++ dataset**, which can be found in **Appendix 6** of the revised manuscript.

---

> > ### Author Response · Authors · 2024-11-25
> > **Appreciation for Review and Request for Feedback**
> >
> > Dear Reviewer 37wm,
> >
> > We want to convey our sincere appreciation for the valuable insights and suggestions you provided regarding our work.
> >
> > We have made efforts to address the concerns and queries you raised during the rebuttal process. It would be immensely helpful to receive feedback on whether our response effectively alleviated any doubts you may have had. Your feedback is crucial to enhancing the quality of our work.
> >
> > Recognizing the demands of your busy schedule, we genuinely appreciate your contribution to the refinement of our manuscript. As the end of the rebuttal period is approaching, we eagerly await your reply before the end.
> >
> > Once again, thank you for your time and consideration.
> >
> > Best regards,
> >
> >  Authors

---

### Official Review · Reviewer_nfD6 · 2024-11-04

**Soundness:** 2
**Presentation:** 2
**Contribution:** 3
**Rating:** 5
**Confidence:** 5

**Summary:**

This paper proposes a new method for deepfake detection based on MLLMs. The method enhances the deepfake detection capabilities of MLLMs by ranking forgery-related features, strengthening strong features and augmenting weak features. In addition, the paper raises the explainability of MLLMs' inference through fine-tuning. Evaluations on multiple datasets prove the effectiveness of the proposed method.

**Strengths:**

1.	The paper proposes Model Feature Assessment(MFA), Strong Feature Strengthening (SFS) and Weak Feature Supplementing (WFS) to enhance the detection capabilities of MLLMs, contributing to MLLM-based algorithms on deepfake detection.

2.	The paper points out the limitations of pretrained MLLMs and introduces a pipeline for constructing datasets based on more effective forgery features.

3.	The experiments provide different protocols and a wide range of evaluation to prove the effectiveness of the method.

**Weaknesses:**

1.	The authors claim that human-annotated VQA data (zhang et al, 2024) may not be ideal as standard answers for fine-tuning MLLMs. Instead, they propose using MLLM-generated data to fine-tune MLLMs. This raises an important question: if all answers are expressed in human's natural language, how can MLLM-generated data establish a more 'standard' answer? Even in the rebuttal response, the author claims their collection method is efficient instead of being a "standard" answer.

2.	The generated questions lack reliability due to the absence of a human verification process to ensure the accuracy of these fake features. Additionally, the authors use data generated by MLLMs to evaluate the MLLMs themselves, which raises concerns about the validity of this approach. This circular logic is confusing and calls into question the robustness of the evaluation.

3.	The authors claim to study the intrinsic capabilities of MLLMs on detection. However, Strong Feature Strengthening (SFS) and Weak Feature Supplementing (WFS) seem to be a process of dataset construction based on more effective features rated by the ranking in Sec. 3.3, thus becoming less pervasive in how it improves the explainability of MLLMs. Although the rebuttal response emphasizes its effectiveness, I am still not convinced by it. For example, in Fig. 5, what if the blending score is wrong? what if the face is not generated by the blending process so the blending score is low while the face is still fake?

4.	The explanation ability of X^2-DFD should also be quantitatively evaluated in the main paper. The authors mention conducting a human study to assess explanation performance in Fig. 7, but they only verbally claim that their model is superior and provide a single qualitative example in the appendix. This lacks reliability and ignores the main contribution of explainability.

5.  A new question after the rebuttal: Line 176, LLaVA and XceptionNet achieve really unreliable detection performance, e.g.., 63.7% and 75.8%; this means both of them are almost equally incapable in the detection task, and analyzing them is less meaningful.

**Questions:**

1.	In Figure 1, I wonder why X^2-DFD predicts a lower fake probability on the second picture compared with a pre-trained MLLM. Meanwhile, the response of X^2-DFD considers the image completely fake, which is contrary to the predicted probability.

2.	Figure 2 is too small; the numbers are barely visible.

3.	There are undefined and inconsistent expressions. In Sec. 4.2.3 and Figure 5, The abbreviation of WCS is undefined in the aforementioned method. In Sec. 5.4 and Tab. 3, GCS has never been seen before.

4.	More details on the implementation of fine-tuning MLLMs should be provided. Which part of the parameters are trained? The dataset information should be detailed.

[Summary]: I appreciate the author's efforts in the rebuttal, while I still would like to maintain my first-round score and suggest this paper revised for the next submission.

---

> ### Author Response · Authors · 2024-11-23
> **Response to Reviewer nfD6 (Part 1/2)**
>
> **Q1. How can MLLM-generated data establish a more 'standard' answer?**
>
> **A1:** Thank you for pointing out the ambiguity in our use of "standard."
> We agree with the reviewer that the word could lead to a lack of clarity, so we have eliminated this expression and substituted it with **"efficient"**, which we think to be more appropriate and accurate here.
> This is because manually creating detailed annotations by humans can be costly and inefficient, while MLLM-based data generation is **cost-effective, potentially scalable, and faster**, making it an efficient choice for creating larger datasets.
>
> Thank you again for bringing this to our attention. And we have modified the content in Lines 133-134 of the **updated manuscript.**
>
>
> **Q2. (a) The generated questions lack reliability due to the absence of a human verification process to ensure the accuracy of these fake features. (b) The circular logic is confusing and calls into question the robustness of the evaluation.**
>
> **A2:** Thank you for your insightful feedback, which has allowed us to address these important points. Here is our clarification:
>
> 1. **Question Reliability and Human Verification:**
>
>    - First, after ranking the generated questions by LLMs, we have already conducted a human verification to ensure the reliability and accuracy of the fake features. However, please note that most questions are generated quite well without any obvious errors or irrelevant information. So we omit the description in the original manuscript. Following your suggestions, we have added this content to Lines 314-316 of the **update manuscript** for better clarity.
>    - Second, we would like to clarify that we have explicitly required the LLMs to generate questions by our designed prompt. So the generated questions by LLMs are expected to be:
>      - **Clear**, focusing on features that indicate forgery.
>      - **Diverse**, avoiding repetition or ambiguity.
>      - **Broad Coverage**: By adjusting the number of generated numbers, we can achieve wider question coverage.
>
>     As shown in **Table 13 and Table 14**, the generated questions effectively meet our requirements, demonstrating the reliability and accuracy of these features.
>
>
> 2. **About Circular Logic:**
>    - We found from *Wikipedia* that "circular logic" is a logical fallacy in which the reasoner **begins with what they are trying to end with**. In our approach, what we begin with (**questions lists** generated by LLMs) and what we end with (**ranked questions lists** after the MFA module) are **distinct**. Therefore, our approach is not circular logic.
>
> Thank you once again for your valuable comments. We hope our response can address your concerns effectively.
>
>
> **Q3. SFS and WFS seem to be a process of dataset construction based on more effective features rated by the ranking in Sec. 3.3, thus becoming less pervasive in how it improves the explainability of MLLMs. More analyses are expected to verify the contributions of the procedure to the enhancement of MLLMs.**
>
> **A3:** Thank you for your valuable feedback. We improve the explainability by leveraging the stronger discriminative features and supplementing the weak using EDDs. Specifically:
>
>    - **Focusing on Strong Features:** by fine-tuning MLLMs' focus on strong forgery features, we aim to encourage the MLLM to explain using only the strong discriminative features, paying less attention to the weak ones, the MLLM is not "familiar with", much like a feature selection process.
>
>    - **Supplementing the weak using EDDs:** For those features that MLLMs might not be "good at" learning such as the subtle blending artifacts (ranked at the bottom), we improve the explainability by leveraging the accurate information from EDDs and encouraging MLLMs to learn how to use these information for **credible and reliable explanations.**
>    - **Results Analysis:**
>      - **Feature Recognition Enhancement:** As shown in **Figure 6**, the model demonstrates significant improvements in recognizing strong features, with a **28.6% improvement** for the Top 1/3 features and **14.6% improvement** for the Top 1/2 features. This enhancement in feature recognition has also improved the model's overall explainability, making its explanations more sound and reliable.
>      - **Human Preference Improvement:** **Figure 7** highlights a notable increase in **human preference ratings** for the model's explanations.
>
> Thank you once again for your insightful comments.

---

> ### Author Response · Authors · 2024-11-23
> **Response to Reviewer nfD6 (Part 2/2)**
>
> **Q4. The explanation ability of X^2-DFD should also be quantitatively evaluated in the main paper. The authors mention conducting a human study to assess explanation performance in Fig. 7, but they only verbally claim that their model is superior and provide a single qualitative example in the appendix.**
>
> **A4:** Thank you for your thoughtful feedback. Below, we clarify and address your concerns:
>
> 1. **Challenges in Quantitatively Evaluating the Explainability in the Field:**
>    - We would like to clarify that there is a lack of **detailed annotations of specific forgery artifacts**, making it difficult to accurately quantify and evaluate explainability. To our knowledge, we have not found any existing studies that propose such quantitive evaluation metrics for explainability that can be directly used in our framework. If we have overlooked any related work, we kindly ask the reviewer to remind us and point it out.
>
>
> 2. **Reliability of Our Human Study:**
>    - The lack of detailed annotations limits the ability to accurately quantify explainability, but human evaluators can also offer a practical solution for **quantitive assessment**, as detailed in **Section 5.5** and **Appendix A.4**:
>    - Although the human assessment could not be completely objective, our human study can still be **reliable** for evaluation, as follows:
>      - Our human study included **well-educated participants** who were provided with **clear guidelines** before the experiment to ensure they could deliver reliable results.
>       - We have added additional experimental details about the human study, highlighted in **red** in **Appendix A.4: Human Study Details and Analysis**(lines 928-933). Due to the double-blind review process, more specific details, including **ethical approval documents with official stamps**, will be included in the final version.
>       - We believe these considerations and designs can provide a reliable quantitive assessment of the explainability, though there is no accurate and detailed annotations of specific forgery artifacts available.
>
> Thank you once again for your valuable comments, which have been instrumental in enhancing the clarity and rigor of our work.
>
>
> **Q5: The issues in the paper might cause misunderstanding :
>     1. Figure 1, a lower fake probability on the second picture compared with a pre-trained MLLM.
>     2. Figure 2 is too small, the numbers are barely visible.
>     3. Undefined and inconsistent expressions. WCS and GCS in paper**
>
> **A5:** Thank you for carefully reviewing our paper and pointing out details that may cause misunderstanding. We have revised all the mentioned content in the updated manuscript to address these issues:
>
> - **Figure 1**: Update **probability** from `0.37` to `0.96` for improved accuracy. (lines 62-63)
> - **Figure 2**: Enlarge the font to ensure the numbers are clearly visible. (lines 184-187)
> - **Section 4.2.3 and Figure 5**: (lines 358-366/478-479/492-493)
>   - Unifiy **WCS (Weak Capability Strengthening)** to **WFS (Weak Feature Strengthening)**.
>   - Unifiy **GCS (Good Capability Strengthening)** to **SFS (Strong Feature Strengthening)**.
>     These terms were standardized to ensure consistency and improve readability.
>
> Thank you once again for your valuable suggestions, which have greatly helped us enhance the clarity of our work.
>
>
>
> **Q6. More details on the implementation of fine-tuning MLLMs should be provided. Which part of the parameters are trained? The dataset information should be detailed.**
>
> **R6:** Thank you for requesting additional details on our fine-tuning approach. Below, we provide clarification on the fine-tuning methodology and dataset information:
>
> - **Fine-Tuning Methodology**:
>   - The fine-tuning of MLLMs is conduct in **Section 4.2.2 Step 3: MLLM Fine-Tuning**. Specifically, we modify the **projector module** and apply LoRA fine-tuning on the **language model**.
>
> - **Dataset Information**:
>   - **Image Data**: In addition to the datasets mentioned in **Section 5 Experimental Settings**, we adhere to the standard deepfake benchmark (**DeepfakeBench [1]**) for processing the original data. Face alignment and face cropping procedures are consistent with DeepfakeBench, and we use **eight frames per video** for model training.
>   - **VQA Data**: The VQA dataset is built through **two rounds of dataset construction** (**SFS**, as shown in **Figure 4**, and **WFS**, as shown in **Figure 5**) and one round of **VQA-specific fine-tuning**. Additionally, the annotation for **WFS** in **Figure 5** has been revised to improve clarity and reader comprehension(lines 351-353/358-359).
>
> Thank you once again for your thoughtful feedback, which has helped us refine and clarify our methodology.
>
>
>
> REFERENCE:
>
> [1] Yan Z, Zhang Y, Yuan X, et al. Deepfakebench: A comprehensive benchmark of deepfake detection. NeurIPS 2023.

---

> > ### Author Response · Authors · 2024-11-25
> > **Appreciation for Review and Request for Feedback**
> >
> > Dear Reviewer nfD6,
> >
> > We want to convey our sincere appreciation for the valuable insights and suggestions you provided regarding our work.
> >
> > We have made efforts to address the concerns and queries you raised during the rebuttal process. It would be immensely helpful to receive feedback on whether our response effectively alleviated any doubts you may have had. Your feedback is crucial to enhancing the quality of our work.
> >
> > Recognizing the demands of your busy schedule, we genuinely appreciate your contribution to the refinement of our manuscript. As the end of the rebuttal period is approaching, we eagerly await your reply before the end.
> >
> > Once again, thank you for your time and consideration.
> >
> > Best regards,
> >
> >  Authors

---

### Meta-Review · Area_Chair_MNFA · 2024-12-17

**Metareview:**

This paper proposes a MLLM-based method to detect face forgery. All reviewers gave positive recognitions to adopting MLLMs for explainable deepfake detection. However, part of the reviewers still had concerns about experiments, opinions and writing. I have reviewed this revised paper and had some concerns that were consistent with reviewers.

Some necessary experiments or analyses are missing to support the authors' views or conclusions. For example, the performances on the low-quality images are not evaluated (e.g., using the model trained on the RAW version of FF++ to test the FF++ with the C23 or C40 version), the discussion on whether LLM would misunderstand normal post-processing traces (e.g., blur or contrast) as forgery traces is not provided, specific experimental results are lacking to demonstrate that applying SFS to weak features is ineffective and that the fine-tuning results are similar to the No SFS baseline, sufficient quantitative results are missing to support the claimed enhancements to explanation of the pre-trained MLLM.

In addition, given that the large gap between the Phi-3-vision and the LLaVa-7B model, the stated view that the framework does not rely on a specific MLLM is not sufficiently convincing. According to this view, the proposed method should be able to achieve similar detection performance under different MLLMs. Besides, some obvious writing issues remain in the revised version, e.g., missing citations in L916 and L1115. Based on the above considerations, I do not recommend accepting this paper.

**Additional Comments On Reviewer Discussion:**

The authors provided rebuttals for each reviewer. Reviewer FRF6 provided a response but still had concerns about the technical depth and contribution. Considering that other reviewers did not provide further responses and that the rating scores varied widely, I reviewed this revised paper and had concerns about experiments, opinions and writing, which were also raised by reviewer nfD6, 37wm, Umwy and FRF6.

---

### Decision · Program_Chairs · 2025-01-22

Reject